# AutoMiSeg: Automatic Medical Image Segmentation via Test-Time Adaptation of Foundation Models

## Abstract

Medical image segmentation is vital for clinical diagnosis, yet current deep learning methods often demand extensive expert effort, i.e., either through annotating large training datasets or providing prompts at inference time for each new case. This paper introduces a zero-shot and automatic segmentation pipeline that combines off-the-shelf vision-language and segmentation foundation models. Given a medical image and a task definition (e.g., "segment the optic disc in an eye fundus image"), our method uses a grounding model to generate an initial bounding box, followed by a visual prompt boosting module that enhance the prompts, which are then processed by a promptable segmentation model to produce the final mask. To address the challenges of domain gap and result verification, we introduce a test-time adaptation framework featuring a set of learnable adaptors that align the medical inputs with foundation model representations. Its hyperparameters are optimized via Bayesian Optimization, guided by a proxy validation model without requiring ground-truth labels. Our pipeline offers an annotation-efficient and scalable solution for zero-shot medical image segmentation across diverse tasks. Our pipeline is evaluated on seven diverse medical imaging datasets and shows promising results. By proper decomposition and test-time adaptation, our fully automatic pipeline not only substantially surpasses the previously best-performing method, yielding a 69% relative improvement in accuracy (Dice Score from 42.53 to 71.81), but also performs competitively with weakly-prompted interactive foundation models.

## 1 Introduction

Medical image segmentation plays a critical role in diagnosis and treatment planning (Patil & Deore, 2013; Litjens et al., 2017). Artificial Intelligence has emerged as a transformative force in this domain, significantly enhancing the efficiency and accuracy of clinical workflows (Hesamian et al., 2019). In particular, deep learning-based segmentation models have outperformed traditional computer vision techniques by leveraging large datasets and effective deep feature learning (Ronneberger et al., 2015a; Zhou et al., 2018; Oktay et al., 2018). Despite these advancements, most state-of-the-art models rely heavily on supervised learning, which requires extensive and high-quality annotations provided by medical experts (Papandreou et al., 2015). Furthermore, supervised models have poor scalability as they are constrained to the pre-defined classes and image domains supported by the training data.

In recent years, the emergence of vision foundation models like the Segment Anything Model (SAM) (Kirillov et al., 2023) offers promising new avenues for more efficient image segmentation. Unlike conventional methods that require separate training on each individual dataset, SAM enables general segmentation at inference time using simple prompts such as points or bounding boxes, significantly reducing the need for extensive labeled data. More advanced models such as MedSAM (Ma et al., 2023) and SAM-Med2D (Cheng et al., 2023) further improve medical image segmentation by fine-tuning SAM on clinical data. However, they may still fail on certain datasets due to the wide variability in medical image modalities, low-contrast features, and subtle anatomical boundaries. Moreover, inference-time prompting still requires expert supervision, which becomes non-negligible when processing large batches of data.

Figure 1: Conceptually illustration of our proposed **annotation-free** and **non-interactive** paradigm for medical image segmentation (AutoMiSeg), compared with traditional supervised and interactive segmentation, which require considerable efforts of medical experts.

The goal of this paper is to realize a brand new paradigm of automatic medical image segmentation that embodies both the *zero-shot* capacity of the prompting paradigm and the *annotation-free inference* characteristic of the supervised learning paradigm. The idea is illustrated in Figure 1. To achieve this goal, the following key challenges must be addressed:

(1) Medical image segmentation is too complex for a single existing model to achieve effective zero-shot and annotation-free prediction. Therefore, properly decomposing the task into a sequence of simpler, modular steps is crucial for building a robust and adaptable pipeline.

(2) As medical images often exhibit domain-specific characteristics that differ from the data used to train general-purpose foundation models, effectively adapting medical images to the pre-trained models without losing their generalization capacity is essential for the broad applicability across various datasets.

This paper presents a general pipeline **AutoMiSeg** for zero-shot, annotation-free automatic medical image segmentation. Given a medical image and a task description (e.g., "segment the optic disc from an eye fundus image"), the pipeline automatically generates the corresponding target mask without additional training or manual annotations. Our basic pipeline consists of a grounding module which aims to provide spatial prompts and an promptable segmentation module for the final mask generation. A prompt boosting module is integrated between them to enhance the prompt quality. Based on the pipeline, we propose a novel test-time adaptation method which employs Bayesian Optimization on a set of Learnable Test-time Adaptors (LTAs). A surrogate validation model is designed to evaluate the segmentation output, and its feedback is used to optimize the LTAs.

Experiments across seven diverse medical imaging datasets demonstrate the effectiveness of AutoMiSeg. Our method delivers a 69% relative improvement in accuracy, raising the average Dice Score from 42.53 up to 71.81, which represents a practically significant advancement over the strongest prior solution. It also achieves competitive performance compared to weak-prompt foundation models, highlighting the potential of the new automatic segmentation paradigm.

The significance of our work can be highlighted from the following aspects.

- Our work is among the first to focus on self-adapting (rather than directly integrating) a sequence of foundation models in medical imaging settings.

- We introduce the first TTA-based pipeline for interactive medical image segmentation, addressing the domain gap between specialized medical domains and a foundation model pipeline.

- Our TTA is specifically developed for non-differentiable foundation model pipelines. It fundamentally differs from traditional TTA methods which apply to standard supervised models.

## 2 RELATED WORK

**Medical Image Segmentation.** Deep learning, particularly U-Net architectures (Ronneberger et al., 2015a; Zhou et al., 2018; Oktay et al., 2018), has become the standard for medical image segmentation but is often hampered by the need for large, manually annotated datasets (Hesamian et al., 2019). To reduce this dependency, researchers have explored few-shot (Snell et al., 2017),

weakly-supervised (Papandreou et al., 2015), and zero-shot learning (ZSL) methods (Xian et al., 2019; Xu et al., 2023; Yang et al., 2024). Our work aligns with the emerging trend of *training-free* methods that leverage large pre-trained foundation models directly for downstream tasks through prompting and composition, avoiding task-specific fine-tuning (Kirillov et al., 2023; Bommasani et al., 2021).

**Promptable Models for Medical Image Segmentation.** The Segment Anything Model (SAM) (Kirillov et al., 2023) initiated a paradigm shift towards promptable segmentation. While its zero-shot capabilities are impressive, studies in medical imaging found its performance inconsistent, often requiring precise user prompts and struggling with fine details or low-contrast regions (Mazurowski et al., 2023; He et al., 2023; Deng et al., 2023). This led to medical-specific adaptations like MedSAM (Ma et al., 2023). Besides SAM based pipelines, One-Prompt (Wu & Xu, 2024) developed a customized network structure that accepts various types of prompt, such as doddle, box, click, etc, for medical image segmentation.

A key challenge remains the automation of prompt generation. Recent works have built text-driven pipelines, such as SaLIP (Aleem et al., 2024), TV-SAM (Jiang et al., 2024), TTCS (Chen et al., 2024), MedCLIP-SAM (Koleilat et al., 2024b;a) and others aiming for universal text-prompted segmentation (Zhao et al., 2023). In addition, SimSAM (Towle et al., 2024) simulates user interactions to automatically select informative clicks for SAM. MedKlip (Wu et al., 2023) utilizes both images and medical reports to enhance feature learning in segmentation tasks. In parallel, large-scale supervised models like BiomedParse (Zhao et al., 2025) achieve strong performance on a wide range of tasks but may lack the flexibility of zero-shot approaches for out-of-domain data. Our work is distinct in its fully automatic, training-free, and compositional design, which uses a dedicated grounding model to interpret text queries for initial localization, followed by a segmentation model.

**Test-Time Adaptation.** To handle domain shifts without retraining, Test-Time Adaptation (TTA) adapts models to new, unlabeled data during inference (Wang et al., 2020; Sun et al., 2020). Recent approaches (Farina et al., 2024; Shin & Kim, 2024; Hoang et al., 2024) demonstrated effectiveness in adapting single backbones under distribution shift. However, most existing solutions rely on online backpropagation and hence are not suitable for non-differentiable pipelines. Such approaches are not directly applicable to promptable segmentation pipelines in which multiple pre-trained components (e.g., grounding, prompt boosting, segmentation) interact and are kept frozen. The most relevant work is SaLIP (Aleem et al., 2024), which considers test-time adaptation but still uses a fixed prompting interface and does not address the challenge of domain shift between medical images and general foundation models. Our work pioneers a novel TTA strategy specifically for a compositional segmentation pipeline. We introduce a set of Learnable Test-time Adaptors (LTAs) whose hyperparameters are optimized via Bayesian Optimization, guided by a proxy validation model, making our framework uniquely adaptable in a fully automatic, training-free manner.

**AutoML for Medical Imaging.** A key to the success of our method lies in applying AutoML principles (He et al., 2021) to medical image segmentation. Existing work on AutoML for medical imaging (Jidney et al., 2023; Ali et al., 2024) has been applied to supervised models, focusing on network architecture design (Isensee et al., 2019; Yu et al., 2023a) and hyperparameter configuration for training (Myronenko et al., 2023). Despite boosting segmentation performance, these methods remain limited to task-specific supervised pipelines. To the best of our knowledge, no prior work has considered AutoML in the context of foundation model adaptation. Our work is the first to bridge this gap, offering a new direction towards robust and supervision-free medical imaging solutions.

## 3 METHODOLOGY

### 3.1 OVERVIEW

The AutoMiSeg pipeline achieves text-guided medical image segmentation without requiring any task-specific training or fine-tuning. The pipeline is illustrated in Figure 2.

Let $I \in \mathbb{R}^{H \times W \times C}$ denote a medical image and let $T = \{T_{\text{target}}, T_{\text{whole}}\}$ denote a structured task description, where $T_{\text{target}}$ specifies the object to be segmented (e.g., "optic disc") and $T_{\text{whole}}$ names the image context (e.g., "eye fundus image"). Given $I$ and $T$, AutoMiSeg outputs a binary segmentation

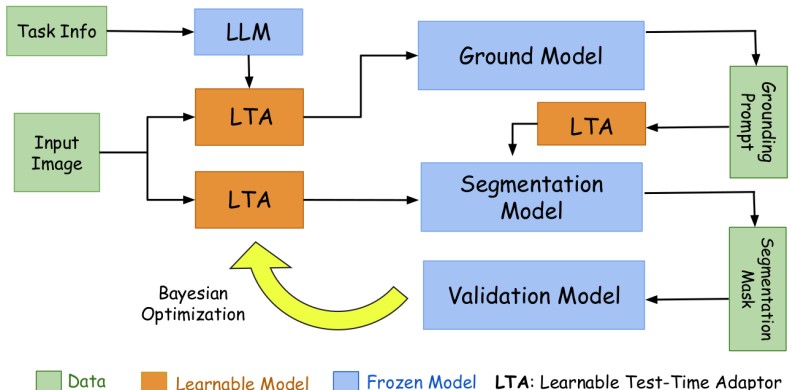

Figure 2: Overview of the proposed AutoMiSeg pipeline. Given a medical image and task description, a grounding module predicts a coarse region, refined by a prompt booster. A promptable segmentation model then generates the binary mask. A validator checks consistency with the task, while Learnable Test-time Adaptors (LTAs) adapt inputs and are optimized via Bayesian Optimization.

mask $M \in \mathbb{R}^{H \times W}$ that matches the task description. The pipeline begins with a **grounding module**, which takes the transformed image $I_G$ and a sentence $S_G$ constructed from the task definition $T$ as input and predicts a bounding box indicating the rough target region. The output is further refined by the **prompt booster**, which improves the prompt's information and quality. The enhanced prompt is then passed to an **promptable segmentation module**, which also takes a transformed image $I_S$ and generates the final segmentation mask $M$.

As a fully automatic pipeline, we need a **validator** to estimate the quality of the output. It's realized by a vision-language model that checks whether the predicted mask $M$ is consistent with the task description $T$. To adapt the general pre-trained models to various medical image domains, we introduce **Learnable Test-time Adaptors (LTAs)**, which consists of a set of tunable operations applied to the inputs of the grounding and segmentation module. We apply **Bayesian Optimization** to search for the optimal configuration of LTAs that maximizes the validator's score.

## 3.2 TARGET AREA GROUNDING

To initially localize the region specified by the text query $T = \{T_{\text{target}}, T_{\text{whole}}\}$, we leverage a pre-trained Vision-Language Model (VLM) capable of text-to-bounding-box grounding. Given the task definition, we first generate a set of prompt sentences $\{S_G | S_G = \mathbf{LLM}(T_{target}, T_{whole})\}$ which describe the visual characteristics of the target in the whole image. Here **LLM** refers to a general large language model which is responsible for generating a diverse set of text descriptions according to its internal medical knowledge about the image type and the target object. The input image $I$ is transformed to $I_G$ by a set of pre-defined vision operations with the specified hyperparameters. A grounding model $\mathrm{M}_{\text{grd}}(I_G, S_G)$ is then employed to generate a bounding box $B = (x_{min}, y_{min}, x_{max}, y_{max})$, which defines the coordinates of the predicted primary region corresponding to $T_{target}$. By default, we use ChatGPT-4o (OpenAI, 2023) as the **LLM** and CogVLM (Wang et al., 2024) as the grounding module. A more detailed introduction of grounding models and CogVLM is presented in Appendix A.

## 3.3 VISUAL PROMPT BOOSTING

Inspired by CoVP (Tang et al., 2024), to refine the initial bounding box $B$, we generate $n$ supplementary positive point prompts via a deterministic, feature-driven process. This utilizes a pre-trained DINOv2 (Oquab et al., 2024), a pretrained vision foundation model valued for its strong semantic features representations. First, an anchor point $p_a$ is set to the geometric center of $B$, and its DINOv2 feature vector $f_a$ is extracted. We then identify the top-$k$ points $P_k$ (e.g., $k = 10$), restricted to lie within $B$ for efficiency, whose DINOv2 features $f(p)$ exhibit the highest cosine similarity to $f_a$. To ensure these $P_k$ points represent diverse parts of the target object, their spatial coordinates are subsequently clustered into $n$ groups (e.g., $n = 3$) using the K-Means algorithm. The centroids of

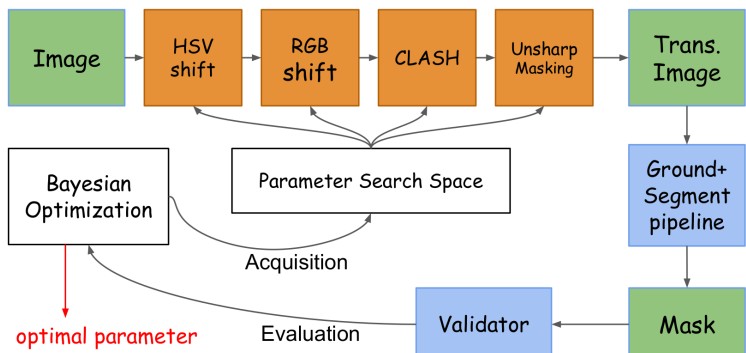

Figure 3: Structural diagram of Learnable Test-time Adaptors (LTAs). While our LTAs involve multiple modules, here we explain the image transformation part as an illustration. The prompt enhancement LTAs are used in prompting the foundation models, but share the same mechanism of BO optimization with image transformation.

these $n$ clusters constitute the final set of generated positive point prompts, $P_c$. These points $P_c$ are then used in conjunction with the original bounding box $B$ by the promptable segmentation model. A more detailed description of the visual prompt boosting module is presented in Appendix B.

### 3.4 PROMPTABLE SEGMENTATION

Equipped with the bounding box $B$ from Section 3.2 and the augmented point set $P_c$ generated in Section 3.3, we utilize a pre-trained promptable segmentation model $M_{seg}$, such as the Segment Anything Model (SAM) (Kirillov et al., 2023). The model takes the image $I$ and the visual prompts $(B, P_c)$ as input and produces a segmentation mask $M$ corresponding to the prompted object, again without any training or fine-tuning.

$$M = M_{seg}(I, B, P_c) \tag{1}$$

### 3.5 LEARNABLE TEST-TIME ADAPTORS

We consider the following two types of tunable variables in our test-time adaptors. A complete description about the variable types and value ranges are introduced in Appendix C. In Figure 3, we illustrate the internal mechanism of the LTA modules, and how they interact with the foundation models and Bayesian Optimization.

**Domain-adapted Image Transformation.** To enhance the compatibility of medical images with general pre-trained vision models, we applied a set of targeted image transformations aimed at improving visual quality and mitigating domain-specific biases. Each transformation was independently applied to both the grounding and segmentation models using separate hyperparameters to optimize performance for their respective tasks. The operations specifically include (1) **HSV shift**, which is used to simulate variations in lighting and acquisition conditions; (2) **Channel-wise RGB shift**, which introduces color perturbations that reduce reliance on fixed intensity distributions often found in medical datasets; (3) **Contrast Limited Adaptive Histogram Equalization (CLAHE)** (Zuiderveld et al., 1994), which is applied to enhance local contrast and reveal subtle anatomical structures typically underrepresented in low-contrast regions; and (4) **unsharp masking**, which is employed to emphasize fine details and edge clarity, which are critical for precise localization and segmentation. These transformations collectively promote better alignment between the test images and the pre-trained vision models.

**Automatic Prompt Enhancement.** LTAs also incorporate an automatic prompt enhancement mechanism to better guide the grounding and segmentation models. Specifically, we consider two tunable choices: (1) selecting the most effective prompt sentence for the grounding model from a set of candidates generated by the LLM, and (2) determining the optimal value of $k$ in the prompt

boosting module, which controls how many points with the most similar features are aggregated to produce the final spatial prompt.

### 3.6 PROXY VALIDATOR

The proxy validator is designed to estimate the chance that a segmentation mask generated by $\mathbf{M}_{seg}$ is the actual target. While it is not a rigorous probability score, we expect that a higher validation score indicates better quality of the generated mask. The designed proxy validator specifically consists of the following two pseudo evaluation tasks.

**Pseudo Evaluation with Zero-shot Classification.** Given an image $I$ and the candidate segmentation mask $M$, we first generate the test image by only keeping the target region as $I_{test} = I \odot (1 - M)$, where $\odot$ denotes pixel-wise multiplication. To design the pseudo classification task, we use a pre-defined template incorporating the task information $T_{target}$ and $T_{whole}$ to prompt a general LLM, and get a list of contrastive categories in the form of text descriptions $\{T_{c_i}\}_{i=1}^m$. We then employ a vision-language model $\mathbf{M}_{val}$ to perform zero-shot classification for $I_{test}$ and output the probability score for $T_{target}$ as the validation score $S_{zc}$.

**Pseudo Evaluation with Image-text Matching** While the zero-shot classification evaluates the image-text alignment with formal medical terminologies, it doesn't consider to verify the segmented region by its expected visual characteristics such as color, shape, textures and so on. Here we employ another pseudo evaluation by considering the image-text matching. Similar to the grounding module, we use a LLM to generate a set of descriptive prompts, and apply the same $\mathbf{M}_{val}$ to calculate the similarity between the test image $I_{test}$ and each descriptor. The average image-text matching score $S_{mt}$ is calculated and contributed to the validation score. The final validation score for an input image $I$ and its candidate segmentation $M$ is $S_{val} = S_{zc} + S_{mt}$. By default, we employ BioMedCLIP (Zhang et al., 2024) as $\mathbf{M}_{val}$. It is worth noting that, with the pre-defined LLM templates, the validation process is fully automatic given the task definition $T = (T_{target}, T_{whole})$ as input. In Appendix D, we present all the LLM templates used in our pipeline.

### 3.7 TEST-TIME ADAPTATION

To make the domain-adapted pipeline compatible to black box modules (e.g., foundation models with API interfaces) and alleviate potential over-fitting, we adopt the Bayesian Optimization (Snoek et al., 2012) in our pipeline for test-time adaptation. The goal is to maximize the evaluation score of the proxy validator by tuning the hyperparameters used in the LTA modules. Specifically, we employ the Tree-structured Parzen Estimator (TPE) as the surrogate model in Bayesian Optimization to better support mixed-type variables. To balance the efficiency and effectiveness, we allow up to $N_t$ trials (e.g., $N_t = 100$) to identify the optimal configuration on a subset of the test set up to $N_s$ (e.g., $N_s = 100$) examples for Bayesian Optimization. The solved configuration is then applied to the entire test set.

Unlike traditional TTA methods that operate on differentiable supervised models by optimizing parameters, our approach is specifically developed for non-differentiable interative-SAM based pipelines. The resulting TTA framework is built upon three core components: 1) *LTAs* that propose candidate masks, 2) *a Proxy Validator* that evaluates their semantic plausibility, and 3) *Bayesian Optimization (BO)* that efficiently searches for the optimal hyperparameter configuration. These components must work jointly to enable zero-shot self-adaptation within the interactive segmentation pipeline.

## 4 EXPERIMENTS

### 4.1 DATASETS

To ensure a comprehensive and diverse evaluation of segmentation performance, we collected seven public medical imaging datasets. These datasets span a broad range of imaging modalities, including fundus photography, endoscopy, ultrasound, dermoscopy and MRI. They cover multiple organ systems

Table 1: Segmentation performance comparison across seven medical imaging datasets. Reported numbers are Dice scores. For supervised baselines except for BiomedParse, we adopt the popular U-Net Ronneberger et al. (2015b) architecture with different backbones such as ResNet (He et al., 2016) and ViTs (Yu et al., 2023b). For AutoMiSeg, we report the average Dice Score and the standard deviations over 3 random experiments. MedSAM results on weak prompts are ignored as they are extremely low.

| Category | Seg model | Kvasir | Busi | Isic2016 | Promise12 | Kidney | SkinCancer | REFUGE | Average |
|----------|-----------|--------|------|----------|-----------|--------|------------|--------|---------|
| **Supervised** | ResNet-18 (He et al., 2016) | 73.9 | 67.4 | 87.8 | 89.5 | 97.9 | 86.4 | 90.4 | 84.8 |
| | ResNet-50 (He et al., 2016) | 69.8 | 63.2 | 88.7 | 88.8 | 97.8 | 84.6 | 90.1 | 83.3 |
| | EfficientNet (Tan & Le, 2019) | 81.2 | 71.1 | 90.3 | 89.2 | 98.1 | 89.0 | 88.2 | 86.7 |
| | MobileNet-v2 (Sandler et al., 2018) | 75.4 | 65.5 | 89.1 | 89.6 | 98.0 | 87.9 | 84.5 | 84.3 |
| | DenseNet-121 (Iandola et al., 2014) | 79.4 | 69.5 | 89.3 | 90.0 | 98.0 | 85.6 | 91.1 | 86.1 |
| | Mix-ViT (Yu et al., 2023b) | 56.9 | – | 89.1 | – | – | 88.1 | 88.1 | – |
| | BiomedParse (Zhao et al., 2025) | 90.7 | 87.4 | 94.3 | 84.5 | 78.9 | 95.6 | 78.5 | 87.1 |
| **Strong prompt** | SAM-Med2D (Cheng et al., 2023) | 89.63 | 89.91 | 93.89 | 87.00 | 87.82 | 92.99 | 83.49 | 89.25 |
| | MedSAM (Ma et al., 2023) | 96.46 | 92.40 | 92.88 | 88.77 | 97.02 | 95.79 | 91.35 | 93.52 |
| | SAM (Kirillov et al., 2023) | 94.83 | 87.41 | 87.38 | 91.10 | 91.97 | 93.59 | 91.89 | 91.17 |
| **Weak prompt** | SAM-Med2D (Cheng et al., 2023) | 61.92 | 76.39 | 87.22 | 62.11 | 56.81 | 88.18 | 52.74 | 69.33 |
| | SAM (Kirillov et al., 2023) | 86.37 | 65.39 | 72.24 | 65.09 | 70.91 | 88.33 | 74.32 | 74.66 |
| **Automatic zero-shot** | SaLIP (Aleem et al., 2024) | 32.35 | 19.39 | 28.05 | 19.21 | 8.15 | 48.66 | 29.06 | 26.41 |
| | MedCLIP-SAM (Koleilat et al., 2024b) | 56.29 | 10.60 | 36.53 | 2.49 | 6.49 | 48.23 | 4.58 | 23.26 |
| | MedCLIP-SAM-v2 (Koleilat et al., 2024a) | 44.57 | 34.26 | 47.33 | 17.15 | 32.55 | 74.52 | 47.35 | 42.53 |
| | AutoMiSeg (**Ours**) | **73.95**±1.55 | **61.68**±1.20 | **70.03**±0.67 | **59.11**±0.86 | **74.25**±1.35 | **85.15**±0.52 | **79.95**±1.08 | **72.02** |

such as the eyes, gastrointestinal tract, breast, skin, prostate and kidneys, enabling an assessment of generalization across heterogeneous domains.

Here we provide a brief introduction of the datasets. (1) The REFUGE (Orlando et al., 2020) dataset includes retinal fundus photographs used to segment the optic disc for glaucoma risk assessment. (2) The Kvasir (Pogorelov et al., 2017)dataset contains endoscopic images of the gastrointestinal tract with annotated polyps. (3) The Busi (Al-Dhabyani et al., 2024) dataset provides ultrasound images of the breast labeled for benign and malignant tumors. (4) The ISIC2016 (Gutman et al., 2016) dataset includes dermoscopic images annotated for general skin lesion segmentation. (5) The UWSkinCancer (Vision and Image Processing Lab, University of Waterloo, 2021) dataset focuses on skin cancer detection from dermoscopic images. (6) The Promise12 (Litjens et al., 2014) dataset comprises T2-weighted prostate MRI scans annotated for prostate volume segmentation. (7) The Usforkidney (Song et al., 2022) (siatsyx, 2024) dataset offers kidney ultrasound images labeled for tumor segmentation. Except for REFUGE (Orlando et al., 2020), all the remaining datasets are from MedSegBench (Kuş & Aydin, 2024) and we adopt the official test splits.

## 4.2 EVALUATION

We evaluate our AutoMiSeg pipeline on the official test split of each dataset to ensure a fair and consistent comparison. As our method represents the first fully automatic, non-interactive zero-shot segmentation framework in the medical imaging domain, we additionally benchmark it against both supervised models trained on pre-defined classes and interactive foundation models guided by various types of prompts. These competing methods serve as practical *upper bounds* for our pipeline, as they rely on different forms of expert intervention, such as labeled data or inference-time guidance, which our approach explicitly avoids.

For supervised models, we present the performance reported in the MedSegBench (Kuş & Aydin, 2024) paper. For interactive models, we follow the practice of SAM-Med2D (Cheng et al., 2023) and One-prompt (Wu & Xu, 2024) to simulate the interaction process. Specifically, we consider two types of interaction, which are strong and weak prompts. The strong prompt is derived from the ground truth mask as the minimal bounding box enclosing all foreground pixels, defined by its top-left and bottom-right coordinates. The weak prompt is defined as a single point uniformly sampled from the target region's foreground pixels. We also evaluate baseline methods SaLIP (Aleem et al., 2024), MedCLIP-SAM (Koleilat et al., 2024b) and MedCLIP-SAM-v2 (Koleilat et al., 2024a), which are also designed to achieve zero-shot medical image segmentation.

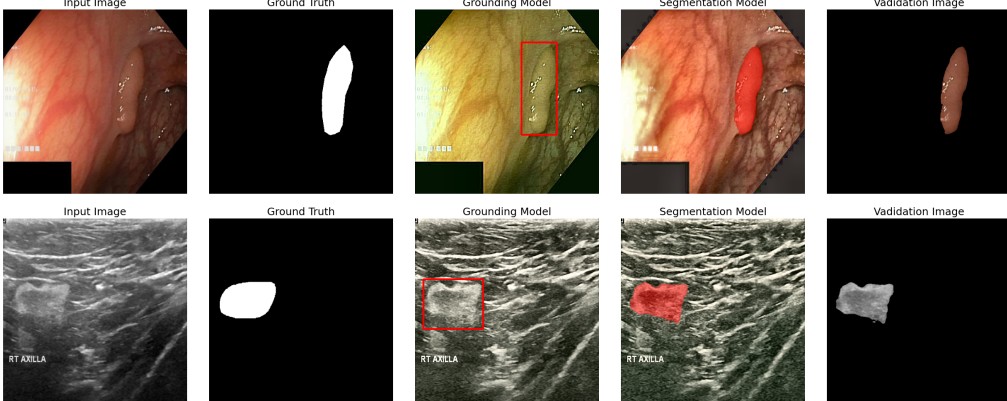

Figure 4: Qualitative visualization of our AutoMiSeg pipeline on two examples from the Kvasir (Pogorelov et al., 2017) and Busi (Al-Dhabyani et al., 2024) dataset. The columns from left to right represent: (1) input image, (2) ground truth, (3) input image of the grounding model and its output, (4) input of the segmentation model and its output, and (5) input of the validator.

## 4.3 MAIN RESULTS

From the results shown in Table 1, supervised models achieve consistently strong results and interactive foundation models with strong prompts demonstrate even higher or comparable performance. However, models using weak prompts exhibit a notable drop in performance. This emphasizes the importance of our proposed strategy that decomposes the segmentation task and incorporates a grounding model to first localize the target region with a bounding box. Within the zero-shot setting group, our method AutoMiSeg substantially surpasses the previously best-performing method MedCLIP-SAM-v2 (Koleilat et al., 2024a) by 69% relative improvement, i.e., improving the average dice score from 42.53 to 71.81. Figure 4 shows qualitative examples of our results. More examples are presented in Appendix E.

Interestingly, within the weak prompt group, SAM (Kirillov et al., 2023) and SAM-Med2D (Cheng et al., 2023) do not exhibit consistent relative performance across all datasets. For instance, SAM (Kirillov et al., 2023) significantly outperforms SAM-Med2D (Cheng et al., 2023) on Kvasir (Pogorelov et al., 2017) (87.21 vs. 61.49), whereas SAM-Med2D (Cheng et al., 2023) yields higher accuracy on Busi (Al-Dhabyani et al., 2024) (77.99 vs. 68.50). These discrepancies suggest that different models (even particularly adapted with medical data such as SAM-Med2D (Cheng et al., 2023)) have dataset-specific strengths and are affected differently by prompt ambiguity, reinforcing the need for test-time adaptation in zero-shot settings.

Our fully automatic pipeline achieves comparable performance with interactive foundation models under weak prompts. While it does not yet reach the performance levels of supervised models or strong-prompt foundation models, this result underscores the promise of the automatic, zero-shot segmentation paradigm[1]. In addition, in many real-world medical scenarios, annotation is costly and expert time is limited, making low-shot learning and Parameter-Efficient Fine-Tuning (PEFT) more realistic supervised baselines. Under such conditions, our zero-shot pipeline demonstrates stronger robustness compared with supervised models. Our pipeline is also shown robust to reasonable levels of natural and artificial noises. These additional experimental results are provided in Appendix F.

By eliminating the need for manual prompts and annotations, our approach provides a scalable solution that balances performance with usability, and demonstrates the potential of further research in this direction. Regarding the efficiency, our method shows comparable inference speed with existing zero-shot pipelines and reasonable test-time training cost involved by Bayesian Optimization. BO can also be accelerated by a warm-starting strategy without hurting the performance. Detailed results are in Appendix G.

---

[1]Note that BiomedParse (Zhao et al., 2025) also supports text input for segmentation. Although this capability resembles a zero-shot paradigm, these approaches still fundamentally fall under the supervised category. A more detailed discussion is provided in Appendix F.

Table 2: Performance comparison across different combinations of $P_{\text{grd}}$ and $P_{\text{seg}}$ on Kvasir (Pogorelov et al., 2017) and Busi (Al-Dhabyani et al., 2024) datasets. In these experiments, we fix the optimal choice of the prompt enhancement hyperparameters and only affect the domain-adapted image transformation process for simplicity. Base refers to using the original image without any transformation.

| $P_{\text{grd}}$ | $P_{\text{seg}}$ | Kvasir (Pogorelov et al., 2017) (endoscopy) | Busi (Al-Dhabyani et al., 2024) (ultrasound) |
|---|---|---|---|
| Optimal | Optimal | 74.80 | 61.65 |
| Optimal | Base | 71.43 | 57.13 |
| Optimal | Random | 66.97 | 54.86 |
| Base | Optimal | 25.94 | 15.72 |
| Random | Optimal | 22.78 | 15.09 |

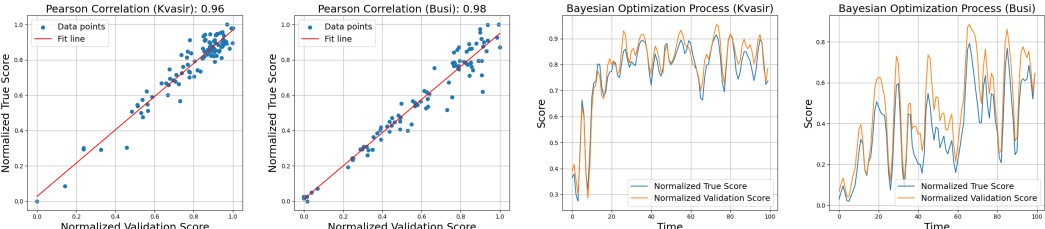

Figure 5: The left two figures show correlation between the normalized validation scores and true scores on the Kvasir (Pogorelov et al., 2017) Busi (Al-Dhabyani et al., 2024) datasets with the searched hyperparameters. The right two figures present the process of Bayesian Optimization.

## 5 MECHANISM ANALYSIS

### 5.1 ROLE OF THE GROUNDING MODEL

We find that the grounding model is the most critical component in the pipeline. This is expected, since the segmentation model relies on the grounding box to localize the target. The results show a dramatic drop in performance when the grounding model uses base or random transformations. For example, on the Kvasir dataset, the Dice score drops from 74.80 (Optimal) to 25.94 (Base) and 22.78 (Random). This confirms that test-time image transformation is essential for grounding accuracy and the reliability of the whole pipeline. When the grounding model is optimal, the segmentation model is more robust to changes. Even with random or no transformation, the performance only drops slightly. This also aligns with the observations in Table 2 that foundation models with strong box prompts achieve excellent performance. Interestingly, random transformations perform worse than using the original image. This suggests that poorly chosen transformations may hurt performance more than help. Careful searching the hyperparameter space is necessary to achieve good results. Beyond the image transformations, text prompt selection in LTAs is also essential to the grounding performance. Detailed analyses are presented in Appendix J.

### 5.2 PERFORMANCE OF THE PROXY VALIDATOR

Additional analyses are presented to verify both the quality of the proxy validator and the dynamics of the Bayesian optimization process. In the analyses, ground truth masks are used post hoc to assess the alignment between the validation scores and the actual performance (measured by Dice score), without influencing the optimization directly. As we can observe in Figure 5, the two left plots reveal a strong linear relationship between the normalized validation scores and true scores on the Kvasir (Pogorelov et al., 2017) and Busi (Al-Dhabyani et al., 2024) datasets, with high Pearson correlation coefficients, confirming that the designed validator serves as a reliable surrogate for model performance. Meanwhile, the two right plots show how the validation and true scores evolves during Bayesian Optimization. The overall trends clearly move toward higher-performing configurations, validating the effectiveness of the search process. Some fluctuations (especially for the Busi (Al-Dhabyani et al., 2024) dataset) reflect the inherent trade-off between exploring new hyperparameter regions and exploiting promising ones.

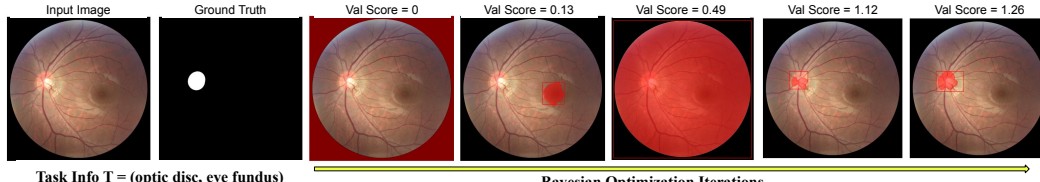

Task Info T = (optic disc, eye fundus) — Bayesian Optimization Iterations

Figure 6: An example from the REFUGE (Orlando et al., 2020) dataset showing how the segmentation quality gradually improves in the optimization process.

Figure 6 illustrates a representative case from the REFUGE (Orlando et al., 2020) dataset, where the task is segmenting the optic disc from a fundus image. The optimization process begins with random configurations in the LTA space, often resulting in the segmentation of background regions. In early stages, the macula, which has a similar shape and size to the optic disc, is mistakenly segmented. However, the validator assigns low zero-shot classification and image-text matching scores to these incorrect predictions. As the optimization progresses, the pipeline gradually refines its focus, ultimately localizing the optic disc and producing accurate segmentation results.

We also find hyperparameter selection with BO on a batch of test samples show better performance than per-sample BO. As shown in Figure 7, although per-sample BO achieves higher validation scores, the overall true scores decrease compared to the default setting of batch BO, as BO tends to overfit each individual test example and may lose the chance to find more robust configurations.

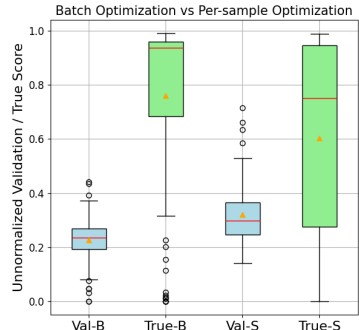

## 6 LIMITATION AND FUTURE WORK

While AutoMiSeg substantially outperforms prior automatic zero-shot segmentation methods, a clear performance gap remains compared to supervised models and foundation models guided by strong expert prompts. Given the modular nature of our pipeline, a promising future direction is to expand the search space to include different model configurations. As shown in our preliminary ablation studies (Appendix H), there is considerable room for improvement through better foundation model selection within the AutoML framework. Our method will also

Figure 7: Evaluate per-sample BO on the Kvasir dataset. B refers to batch and S refers to per-sample.

naturally benefit from the continuous advances of general foundation models. Another important avenue is to develop more effective test-time adaptation strategies tailored for the non-differentiable and compositional nature of the pipeline.

Another challenge lies in the Validator, which tends to verify the recognized regions in isolation and incorporates limited contextual cues. Our current strategy is to use a rectangle that fully covers small segmentation masks, which is practical but only partially alleviates this issue. To further enhance the validation reliability, a promising future direction is to more systematically integrate contextual information. One potential approach is to apply explainable AI techniques to CLIP-based models, inspired by recent advances in weakly supervised segmentation in general computer vision Yang et al. (2025). Another direction is to utilize region-aware CLIP models, which may improve the alignment between text features and patch-level image features Sun et al. (2024). However, these ideas remain challenging to adopt in medical domains due to the lack of high-quality region-aware CLIP models. Exploring these directions represents a valuable avenue for future work.

Another future direction is to design strategies to improve clinical safety. For example, the validation score can be used as an indicator of prediction uncertainty. We can also perform posterior sampling from the BO results to obtain multiple predictions and check their consistency. Since the hyperparameters are mostly related to image transformations, these images can be processed efficiently in parallel.

# 7 CONCLUSION

This paper presents AutoMiSeg, a fully automatic, zero-shot segmentation pipeline for medical images that eliminates the need for manual annotations or interaction at inference time. By leveraging pretrained vision-language and segmentation foundation models, along with a novel test-time adaptation framework, our method effectively addresses the challenges of domain shift and prompt generation in medical image segmentation. Experimental results across seven diverse datasets demonstrate the pipeline's significant advantages over zero-shot baselines and also competitive performance compared to both weak-prompt interactive models. This work underscores the potential of modular, training-free pipelines for scalable medical image analysis and opens up promising directions for future research in improving the performance with more powerful pre-trained models and designing advanced test-time adaptation approaches for more complex pipelines.

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

## A    VISUAL GROUNDING MODELS

Visual Grounding (also known as Referring Expression Comprehension) is a crucial task in computer vision and natural language processing that aims to localize specific objects or regions in an image based on a natural language textual description (a "text prompt" or "referring expression"). Unlike traditional object detection systems that identify objects from a predefined set of categories, visual grounding models are designed to understand free-form, often complex, textual prompts that can describe an object by its attributes, spatial relationships to other objects, or other distinguishing characteristics.

The input to a typical visual grounding model consists of an image and a text prompt indicating the object of interest. The output is usually the set of coordinates for a bounding box that precisely encloses the described object within the image. This capability to "ground" language in visual content makes these models powerful tools for fine-grained image understanding and interaction.

The ability of these models to interpret nuanced language and map it to specific image regions makes them highly valuable for various downstream applications, including human-computer interaction, robotics, image editing, and, as in our work, providing precise visual prompts (bounding boxes) for subsequent processing stages like segmentation. Given their relatively recent surge in capability and application, a brief introduction is warranted for reviewers who may be less familiar with this specific class of models.

For our training-free medical image segmentation pipeline, we required a robust and versatile visual grounding model capable of accurately localizing objects of interest based on textual prompts, which would affect the location and quality of masks generated by the successory mask proposing modules.

We selected **CogVLM** (Wang et al., 2024) as the grounding module in our pipeline for several key reasons:

1. **State-of-the-Art Performance and Open Access:** CogVLM is a powerful, open-source visual language model that has demonstrated strong performance across a wide array of vision-language tasks, including those requiring fine-grained understanding and localization. Its availability facilitates reproducibility and further research.

2. **Effective Grounding Capabilities:** While CogVLM is a general visual language model, its architecture is inherently well-suited for tasks that require grounding textual concepts in images. It can be effectively prompted or adapted to output bounding box coordinates corresponding to objects described in the text. Its ability to handle detailed descriptions and disambiguate objects makes it suitable for generating precise visual prompts.

3. **Strong Generalization and Zero-Shot Potential:** Due to its extensive pre-training on large-scale image-text datasets, CogVLM exhibits impressive generalization capabilities to novel objects and scenarios. This strong zero-shot or few-shot performance aligns perfectly with the "training-free" philosophy of our proposed pipeline, allowing us to leverage its capabilities without task-specific fine-tuning for the grounding step.

## B    VISUAL PROMPT BOOSTING

While the bounding box $B$ provides a coarse localization, promptable segmentation models often benefit from more precise internal points to disambiguate the target object from the background or adjacent structures. We generate these point prompts automatically by analyzing the visual features within the image $I$. This process is deterministic given the chosen pre-trained encoder.

**Anchor Point Selection.**    We define an initial anchor point $p_a = (x_a, y_a)$ as the geometric center of the bounding box $B$.

$$x_a = \frac{x_{min} + x_{max}}{2}, \quad y_a = \frac{y_{min} + y_{max}}{2} \tag{2}$$

**Dense Feature Extraction.**    We employ a pre-trained vision encoder, specifically DINOv2 (Oquab et al., 2024) known for its strong semantic feature representation capabilities without fine-tuning, to extract dense feature maps $F \in \mathbb{R}^{H' \times W' \times D}$ from the image $I$. Let $f(p) \in \mathbb{R}^D$ denote the feature

vector corresponding to a spatial location $p = (x, y)$ in the image (potentially requiring interpolation if $p$ does not align perfectly with the feature grid). Let $f_a = f(p_a)$ be the feature vector of the anchor point.

**Similarity-Based Point Selection.** We identify candidate points within the image (or potentially restricted to the bounding box $B$ for efficiency) that are semantically similar to the anchor point in the DINOv2 feature space. We compute the cosine similarity between the anchor feature $f_a$ and all other features $f(p)$ and select the top-$k$ points $P_k = \{p_1, ..., p_k\}$ with the highest similarity scores:

$$P_k = \underset{p \in I, p \neq p_a}{\text{top-}k} \left( \frac{f_a \cdot f(p)}{\|f_a\| \|f(p)\|} \right) \tag{3}$$

In our experiments, we typically use $k = 10$.

**Point Clustering and Center Calculation.** The top-$k$ similar points might form distinct spatial clusters within the target object. To obtain representative points covering potentially different parts of the object, we cluster the coordinates of the points in $P_k$ into $n$ groups using K-Means clustering.

$$C_1, ..., C_n = \text{KMeans}(\{(x_i, y_i) \mid p_i \in P_k\}, n) \tag{4}$$

where $C_j$ represents the set of points belonging to the $j$-th cluster. We set $n = 3$ in our typical configuration.

**Final Point Prompt Generation.** We calculate the centroid (mean coordinate) $p_{c,j}$ for each cluster $C_j$. These centroids form our set of additional positive point prompts $P_c = \{p_{c,1}, ..., p_{c,n}\}$. The final set of visual prompts for the segmentation model consists of the bounding box $B$ and the generated points $P_c$.

## C LEARNABLE TEST-TIME ADAPTORS

We introduced a set of domain-adapted image transformations within our Learnable Test-time Adaptors (LTAs) in an effort to reduce the domain shift between clinical images and foundation models pretrained on natural images. We applied a series of light-weighted image transformation and prompt enhancement during the test time. The operations are designed to enhance local contrast, simulate modality-specific variability and emphasize the anatomical features that are hard to detect within the original images.

All the transformations have a couple of tunable parameters, and we optimize over them using Bayesian optimization (TPE (Bergstra et al., 2011)). We apply these transformations separately to the grounding input and segmentation input, each with their own set of parameters. We also include two non-augmentation parameters in the search: the grounding prompt index $i_{grd}$ controls which predefined prompt to use for the grounding model, and the number of enhanced prompt points $k$ controls the number of point clusters used to boost the box prompt. These parameters affect the quality and stability of the grounding and segmentation process, especially in noisy or ambiguous settings. A complete description of the involved operations is as follows.

- **HSV Shift:** Shift of image hue, saturation, and brightness. This captures variations in scanner settings or lighting.
- **RGB Shift:** Adds individual offsets to all of the R/G/B channels. This is used to reduce the model's dependence on an even intensity distribution.
- **CLAHE (Zuiderveld et al., 1994):** Performs contrast-limited adaptive histogram equalization for greater visibility in locally low-contrast areas.
- **Unsharp Masking:** Sharpens edges by subtracting a blurred version of the image, making anatomical boundaries clearer.
- **Grounding Prompt Selection:** Controls the choice of a predefined bounding box prompt used for the grounding model. Different prompt configurations can significantly affect alignment between the language and the vision encoder.

Table 3: Hyperparameters and search spaces used in LTA optimization.

| Operation | Hyperparameter | Type | Range | Description |
|---|---|---|---|---|
| HSV Shift | hsv_hue_shift | Integer | [0, 20] | Amount of hue rotation |
| | hsv_sat_shift | Integer | [0, 30] | Change in color saturation |
| | hsv_val_shift | Integer | [0, 30] | Change in image brightness |
| RGB Shift | r_shift | Integer | [0, 20] | Red channel offset |
| | g_shift | Integer | [0, 20] | Green channel offset |
| | b_shift | Integer | [0, 20] | Blue channel offset |
| CLAHE | clahe_clip | Float | [0.0, 4.0] | Clip limit for local contrast |
| | clahe_grid | Integer | [1, 4] | Number of tiles per image axis |
| Unsharp Masking | edge_strength | Float | [0.0, 1.0] | Edge enhancement level |
| Prompt Selection | grd_prompt_id | Categorical | {0,1,...,9} | Choice of grounding prompt ID |
| Prompt Boosting | bst_k_points | Integer | [0, 5] | Number of point prompts |

- **Number of Boosted Points:** Controls the number of points used for box prompt boosting. The choice can be 0, which means prompt boosting is not performed. In practice, we find this component is probably learned to be skipped if the target region contains diverse visual patterns and the central point is not very representative. Otherwise, the prompt boosting module tends to generate a few representative points as additional prompts and make the segmentation boundary more accurate.

We list the hyperparameters and their search spaces below in Table 3. These parameters are optimized using Bayesian optimization with a Tree-structured Parzen Estimator (TPE), maximizing proxy validation scores, and were chosen to provide a moderate set of variations without over-warping the input.

# D PRE-DEFINED LLM TEMPLATES

We use the following templates to generate necessary text resources used in our automatic medical image segmentation pipeline.

## D.1 TEXT PROMPT FOR THE GROUNDING MODEL

---

**Input to ChatGPT-4o**

I want to describe the visual characteristics of the $T_{target}$ in an image of $T_{whole}$. Please generate 10 sentences for the purpose of computer vision annotation. Each sentence should describe the shape, size (relative to the entire image), color, texture, and other visual features that help humans easily recognize the target. Each sentence must begin with "Locate the $T_{target}$ in the image of $T_{whole}$, which". Each sentence should be limited within 20 words. Ensure the 10 sentences are diverse and distinct. Write the results in a .txt file, with each description on a separate line and without line numbers.

---

**Sample Output for the REFUGE dataset**

Locate the optical disc in the image of eye fundus, which is round, bright, and positioned toward the nasal side.
Locate the optical disc in the image of eye fundus, which appears yellowish, slightly elevated, and sharply bordered by retina.
Locate the optical disc in the image of eye fundus, which shows a pale circular region where blood vessels radiate outward.
Locate the optical disc in the image of eye fundus, which is circular, bright pink, and textured with fine vascular structures.
Locate the optical disc in the image of eye fundus, which looks like a smooth light-colored oval near the nasal margin.
Locate the optical disc in the image of eye fundus, which appears as a distinct white spot with central vessel branching.
Locate the optical disc in the image of eye fundus, which displays a soft granular texture and contrasts with darker retina.
Locate the optical disc in the image of eye fundus, which looks bright, circular, and located slightly off-center to the left.
Locate the optical disc in the image of eye fundus, which appears as a light area with radial vascular convergence.
Locate the optical disc in the image of eye fundus, which is smooth, round, pale, and lies near the nasal boundary.

---

## D.2 Generating Contrastive Classes for the Validation Model

> **Input to ChatGPT-4o**
>
> In an image of $T_{whole}$, what are the typical regions that have visual features and positions distinct from the $T_{target}$? List 2 to 4 of the most visually different parts or regions, using names commonly found in medical literature. Write the results in a .txt file, with the names separated by comma without additional output.

> **Sample Output for the REFUGE dataset**
>
> macula, fovea, retinal blood vessels, peripheral retina

In our pipeline, we manually add "background" as a default contrastive class for each dataset in addition to the generated classes. We empirically find this is beneficial to validate the foreground targets with the visual language model $M_{val}$.

## D.3 Generating Target Descriptions for the Validation Model

We use a similar template as that used for the grounding model, only differing in the requirements of prefix and length of results.

> **Input to ChatGPT-4o**
>
> I want to describe the visual characteristics of the $T_{target}$ in an image of $T_{whole}$. Please generate 10 sentences for the purpose of computer vision annotation. Each sentence should describe the shape, size (relative to the entire image), color, texture, and other visual features that help humans easily recognize the target. Each sentence must begin with "It appears", "It looks" or "It is". Ensure the 10 sentences are diverse and distinct. Write the results in a .txt file, with each description on a separate line and without line numbers.

> **Sample Output for the REFUGE dataset**
>
> It appears as a round, pale yellow region slightly off-center, with a smooth texture and well-defined margins that stand out against the darker retina.
> It looks like a circular, elevated area occupying about 10% of the image width, with a pinkish hue and radiating blood vessels.
> It is an oval-shaped, bright region located nasally, with a creamy white color and a central depression known as the optic cup.
> It appears as a clearly demarcated disk with a light peach tone, fine granular texture, and a ring-like structure due to the cup-to-disc contour.
> It looks like a small, reflective area with a yellow-white coloration, prominent vessel branches emerging from its center, and a slightly rough surface.
> It is a slightly oval, pink-colored disk occupying approximately one-eighth of the image, with a smooth edge and a central dip.
> It appears as a bright, circular area with a distinct boundary, a pale central cup, and a surrounding rim that contrasts with the adjacent retina.
> It looks like a sunken, light-toned region, slightly raised at the edges, and serves as the entry and exit point for retinal blood vessels.
> It is a well-circumscribed, yellowish-white disk, relatively small compared to the full image, with a visible vascular pattern originating from its core.
> It appears as a doughnut-like structure with a pink outer ring and a paler inner cup, located toward the nasal side and distinct in texture from the rest of the fundus.

# E MORE QUALITATIVE EXAMPLES

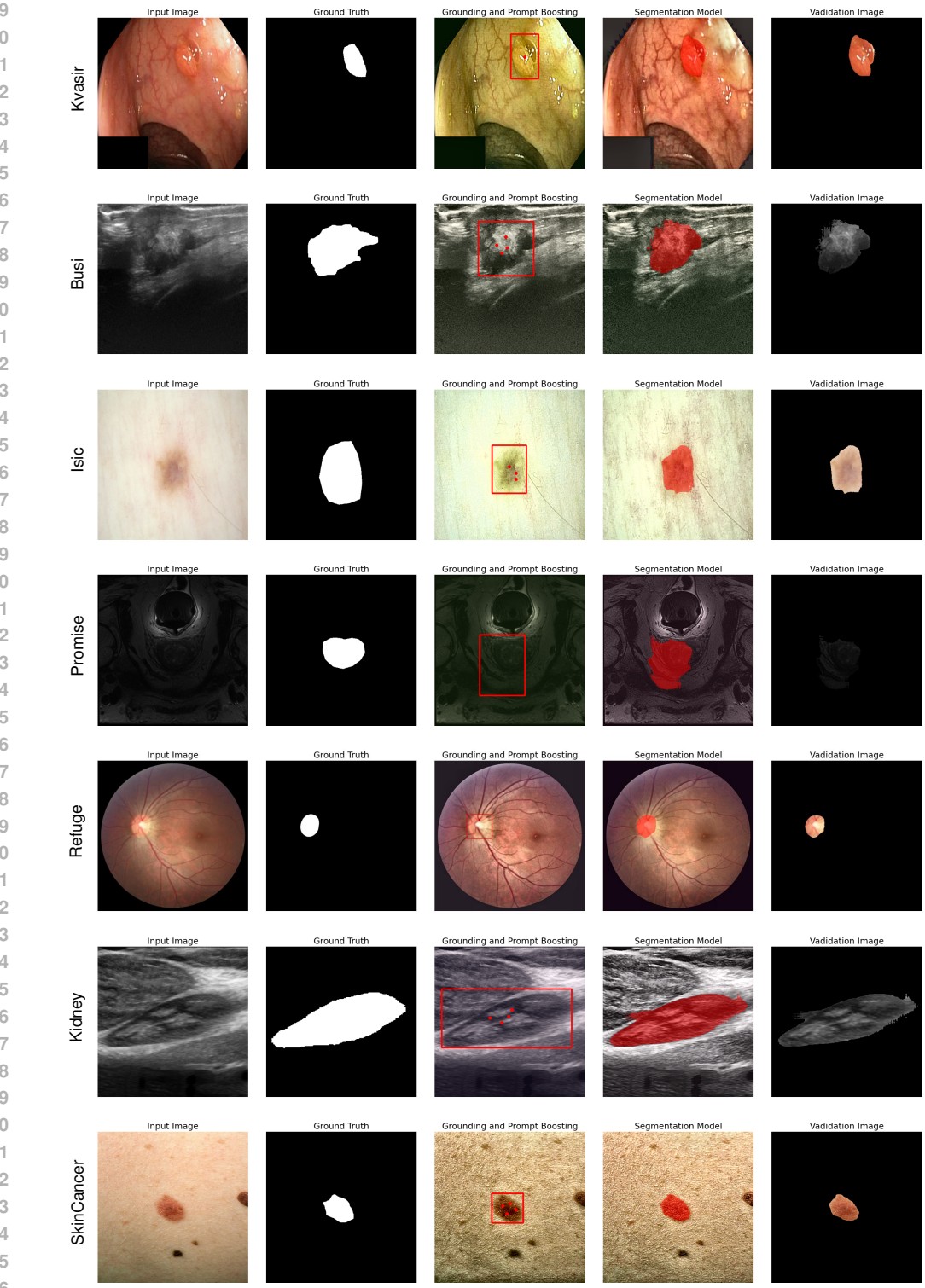

Figure 8: More qualitative results with our automatic segmentation pipeline.

# F  ADDITIONAL EXPERIMENTAL EVALUATIONS

## F.1  COMPARISON WITH TEXT-PROMPTED SUPERVISED MODELS

The recent state-of-the-art medical image segmentation model BiomedParse (Zhao et al., 2025) also accepts text prompts at inference. However, it is not a genuine zero-shot learning model. Its text-conditioning ability is learned in a supervised manner, where the textual inputs during training are largely restricted to domain-specific terms and their limited variations contained in the annotated datasets. Consequently, its zero-shot capability does not extend to unseen domains or novel terminology, and the model exhibits poor generalization on out-of-domain datasets, being inherently constrained by the supervised bias of its training data.

To more comprehensively assess the robustness of our approach and such baselines, we additionally compare performance on datasets outside the domain of BiomedParse (Zhao et al., 2025). The datasets used are the BUU (Lumbar Vertebrae X-ray) dataset (Klinwichit et al., 2023) and the FUSC2021 (Diabetic Foot Ulcer) dataset (Cassidy et al., 2021) . These results have been presented in Table 4. On datasets included in BiomedParse's training data, BiomedParse achieves performance comparable to the evaluated supervised models (as observed in Table 1), as expected. However, on unseen datasets, our approach substantially outperforms BiomedParse as shown in Table 4.

Table 4: Segmentation performance comparison across out-of-domain datasets between BiomedParse and our approach.

| Method | BUU (Lumbar Vertebrae X-ray) | FUSC2021 (Diabetic Foot Ulcer) |
|---|---|---|
| BiomedParse (Zhao et al., 2025) | 4.25 | 21.61 |
| AutoMiSeg (**Ours**) | 67.24 | 56.19 |

## F.2  COMPARISON WITH PARAMETER-EFFICIENT FINE-TUNING MODELS

Here we consider the popular Parameter-efficient Fine-tuning (PEFT) method Low-Rank Adaptation (LoRA) Hu et al. (2022), which also belong to the supervised learning paradigm. To provide a fair comparison, we evaluated LoRA on two representative datasets, Kvasir and BUSI, by fine-tuning the pre-trained SAM model with LoRA (rank = 8), resulting in 0.53% trainable parameters. For LoRA, we considered two settings using either the full training set or only 10% of the training examples. The results show that LoRA's performance is highly dependent on the amount of labeled data: when using the full datasets (700 images for Kvasir and 452 images for BUSI), LoRA surpasses our zero-shot AutoMiSeg; however, under low-data conditions (70 and 45 images, respectively), LoRA does not outperform our zero-shot pipeline. These findings highlight the inherent data-hungry nature of supervised approaches (even for PEFT methods like LoRA), whereas our zero-shot method requires no training data and remains effective in low-shot scenarios.

Regarding the training cost, our method is slightly heavier than the full-data LoRA. This is reasonable because LoRA is a supervised learning method that achieves optimization using *ground-truth labels*, whereas zero-shot pipelines require more exploration with proxy validation using only *unlabeled test data*. Moreover, our BO efficiency could be further improved by warm-starting strategies (see Appendix G.2).

Table 5: Segmentation performance and training cost comparison between LoRA and our approach.

| Data | AutoMiSeg (zero-shot) | LoRA (full data, 15 epochs) | LoRA (10% data, 30 epochs) |
|---|---|---|---|
| Kvasir | 74.80/4.9hrs | 81.38/3.2hrs | 66.59/0.6hrs |
| Busi | 61.65/4.9hrs | 72.31/2.1hrs | 54.71/0.4hrs |

## F.3  ROBUSTNESS TO NOISY INPUT

It is worth noting that robustness to noisy inputs is important for clinical applicability. The medical datasets we evaluate already contain substantial real-world artifacts (e.g., speckle, blur, illumination

variations). Nevertheless, to further validate robustness under simulated perturbations, we additionally injected Gaussian noise into the test images and re-evaluated AutoMiSeg. As shown below, the performance decreases only moderately under 5% and 20% noise, demonstrating that the pipeline remains stable under considerable noise levels, while the Ultrasound images (Busi) show higher robustness to Gaussian noise than Endoscopy images (Kvasir).

Table 6: Segmentation performance across different noise levels.

| Data | clean | 5% noise | 20% noise |
|------|-------|----------|-----------|
| Kvasir | 74.80 | 73.76 | 63.56 |
| Busi | 61.65 | 61.04 | 59.45 |

## G    TRAINING AND INFERENCE COST

### G.1    COMPARISON WITH BASELINES

Here we compare the inference times of the different zero-shot segmentation methods with our approach. All inferences were run on an Nvidia A100 GPU, and the average per-sample inference time (in seconds) over 100 samples from the Busi dataset was computed. These values are presented in Table 7.

Although our pipeline does introduce more components to address the challenging zero-shot problem in medical domains, many of these components are only involved in the one-time adaptation process. Only the grounding and segmentation models are used during inference. As seen in Table 7, our approach achieves a comparable inference speed to the baselines.

Regarding training cost, the BO process in our AutoMiSeg with 100 iterations takes about 5 hours on an NVIDIA A100 GPU, which is comparable to a typical supervised training process (e.g., full fine-tuning or LoRA-based fine-tuning) on SAM. Considering the annotation-free zero-shot setting and our superior segmentation accuracy over the baselines, the overall computational cost is reasonable.

Table 7: Comparison of average per-sample inference times over 100 samples from the Busi (Al-Dhabyani et al., 2024) dataset between other zero-shot segmentation baselines and our approach. All values are in seconds.

| SaLIP (Aleem et al., 2024) | MedCLIP-SAM (Koleilat et al., 2024b) | MedCLIP-SAM-v2 (Koleilat et al., 2024a) | AutoMiSeg (**Ours**) |
|---|---|---|---|
| 1.50 | 0.49 | 2.09 | 1.56 |

### G.2    ACCELERATING BAYESIAN OPTIMIZATION

It is feasible to accelerate BO through few-shot warm-starting. Specifically, we use 10% of the test examples for the first 80 iterations as warm-starting, and then use the full test set in the remaining 20 iterations for deeper exploitation. Specifically, we randomly select 10 test examples in each iteration during the warm-starting phase to ensure diversity of test data used in BO. In this way, the overall BO cost is reduced by more than 70%, while our experiments on REFUGE show that the accuracy remains almost unchanged. We further increased the number of BO iterations using the full test set and observed higher accuracy than the default BO setting, without increasing the overall computation. Therefore, a proper warm-starting (WS) strategy is indeed beneficial for both efficiency and effectiveness.

| BO Strategy | Computation Cost | Dice Score |
|-------------|------------------|------------|
| 100 iters of full-data (default) | 100% | 79.78 |
| 80 iters of WS + 20 iters of full-data | 28% | 79.67 |
| 80 iters of WS + 50 iters of full-data | 58% | 82.50 |

Table 8: Performance and cost comparison of Bayesian Optimization strategies.

## H  MODULAR ABLATION STUDY

We perform an ablation study to evaluate our choice of choosing the base variant of CogVLM as our grounding model for target area grounding. We replace CogVLM with the base variant of Grounding Dino (Liu et al., 2024), while keeping all other components of the pipeline fixed. We evaluate the performance on the Kvasir and Busi datasets, using the same grounding prompts that were used with CogVLM. Since Grounding Dino outputs a number of bounding boxes along with their scores, we select the bounding box with the top score. Similarly, we also evaluate suitable variants for the segmentation and validator modules. For segmentation, we replace the default SAM with MedSAM and SAM-Med2D. For the validator, we replace the default BiomedCLIP with BioVIL-T (Bannur et al., 2023). The results of the ablation study are presented in Table 9.

Table 9: Segmentation performance under ablation when evaluating CogVLM and Grounding DINO separately as grounding models on Kvasir (Pogorelov et al., 2017) and Busi (Al-Dhabyani et al., 2024) datasets.

| Module | Ablation | Kvasir (endoscopy) | Busi (ultrasound) |
|---|---|---|---|
| - | AutoMiSeg default | 74.80 | 61.65 |
| Grounding | CogVLM → Grounding Dino | 67.80 | 51.50 |
| Segmentation | SAM → MedSAM | 78.50 | 66.81 |
|  | SAM → SAM-Med2D | 72.88 | 66.30 |
| Validator | BiomedCLIP → BioVIL-T | 73.60 | 60.65 |

## I  ADDITIONAL EVALUATION ON MEDICAL IMAGE SEGMENTATION DATASETS

To further confirm the general feasibility and robustness of our automatic pipeline, we expand the coverage of imaging modalities and anatomical regions. The newly incorporated datasets include cc359-GE1.5 (MRI, brain) (Souza et al., 2018) , Covid19 (X-ray, lungs) (Rahman et al., 2021), CHAOS (CT, liver) (Kavur et al., 2021), WBC (Microscopy, nucleus) (Zheng et al., 2018), YeaZ (Microscopy, cells) (Dietler et al., 2020), and RoboTool (Endoscopy, surgical tools) (Garcia-Peraza-Herrera et al., 2021). As summarized in the following table, the results demonstrate that our pipeline consistently performs well across this broader spectrum of medical imaging tasks. Specifically, AutoMiSeg significantly outperforms the zero-shot baselines and achieves performance comparable to the weak-prompted interactive SAM (WP-SAM).

Table 10: Segmentation performance comparison across six medical imaging datasets. Reported numbers are Dice scores.

| Category | Seg model | GE1.5 | Covid19 | CHAOS | wbc | yeaz | Robotool | Average |
|---|---|---|---|---|---|---|---|---|
| Supervised | UNet (ResNet-50) | 92.7 | 99.1 | 97.6 | 96.2 | 93.5 | 87.4 | 94.42 |
| Strong prompt | SAM (Kirillov et al., 2023) | 94.42 | 78.90 | 91.20 | 89.52 | 92.56 | 86.00 | 88.10 |
| Weak prompt | SAM (Kirillov et al., 2023) | 87.92 | 73.94 | 71.70 | 81.09 | 89.25 | 76.03 | 79.99 |
| Automatic zero-shot | SaLIP (Aleem et al., 2024) | 71.37 | 73.16 | 33.56 | 49.23 | 69.09 | 44.34 | 56.79 |
|  | MedCLIP-SAM (Koleilat et al., 2024b) | 70.02 | 43.61 | 62.96 | 67.24 | 59.76 | 50.37 | 59.33 |
|  | MedCLIP-SAM-v2 (Koleilat et al., 2024a) | 63.52 | 54.06 | 32.93 | 72.66 | 65.07 | 29.70 | 52.99 |
|  | AutoMiSeg (**Ours**) | **74.80** | **61.65** | **68.38** | **60.61** | **73.05** | **84.41** | **79.78** |

The datasets used in the main paper and appendix can be summarized as follows.

| Dataset | Imaging Type | Anatomy | Target |
|---------|--------------|---------|--------|
| REFUGE (Orlando et al., 2020) | Fundus | Optic Disc | Disc |
| Kvasir (Pogorelov et al., 2017) | Endoscopy | GI Tract | Polyp |
| BUSI (Al-Dhabyani et al., 2024) | Ultrasound | Breast | Tumor |
| ISIC2016 (Gutman et al., 2016) | Dermoscopy | Skin | Lesion |
| UWSkinCancer (Vision and Image Processing Lab, University of Waterloo, 2021) | Dermoscopy | Skin | Cancer |
| Promise12 (Litjens et al., 2014) | MRI | Prostate | Prostate |
| USForKidney (Song et al., 2022) | Ultrasound | Kidney | Tumor |
| cc359-GE1.5 (Souza et al., 2018) | MRI | Brain | Brain |
| Covid19 X-ray (Rahman et al., 2021) | X-ray | Lungs | Lungs |
| CHAOS (Kavur et al., 2021) | CT | Liver | Liver |
| WBC (Zheng et al., 2018) | Microscopy | Blood Cells | Nucleus |
| YeaZ (Dietler et al., 2020) | Microscopy | Yeast Cells | Cell |
| RoboTool (Garcia-Peraza-Herrera et al., 2021) | Endoscopy | Surgical Scene | Tools |

Table 11: Summary of datasets used in our experiments, covering diverse imaging modalities, anatomical regions, and recognition targets.

## J    ANALYSIS ON TEXT PROMPTS

The grounding model is sensitive to both the length and wording of text prompts. Since current grounding models handle short prompts better, we constrain CogVLM prompts to within 20 words. We also observe that wording substantially affects segmentation. As shown below, replacing the optimal prompt with a random one may noticeably change performance on REFUGE:

| Optimal | Random1 | Random2 | Random3 | Random4 |
|---------|---------|---------|---------|---------|
| 79.78 | 69.16 | 67.05 | 78.10 | 72.68 |

Table 12: Effect of using optimal vs. randomly worded prompts on REFUGE.

To address this, our pipeline performs automated prompt search rather than relying on manually designed prompts. An LLM generates candidate prompts using only the imaging modality (e.g., X-ray) and target (e.g., lungs), and BO selects the best prompt (the prompt index is an optimized variable). The LLM templates (Appendix D.1) are general and require no external medical knowledge. The number of text prompts is a hyperparameter. We fix it to 10. Using REFUGE, we find that too few prompts can be suboptimal, while too many may harm BO efficiency as shown in the talbe. We believe this is because more prompts enlarge the search space and may distract BO, while too few reduce the chance of selecting a good prompt.

| n=5 | n=10 | n=20 | n=30 |
|-----|------|------|------|
| 76.88 | 79.78 | 80.67 | 79.21 |

Table 13: Influence of the number of candidate prompts on REFUGE.

