# OpenReview forum: "AutoMiSeg: Automatic Medical Image Segmentation via Test-Time Adaptation of Foundation Models"
_ICLR.cc/2026/Conference — ICLR 2026 Conference Desk Rejected Submission_

### Official Review · Reviewer_t9YK · 2025-10-26

**Soundness:** 3
**Presentation:** 3
**Contribution:** 3
**Rating:** 6
**Confidence:** 4

**Summary:**

The paper proposes a prompt-guided semantic segmentation framework that adapts generic pre-trained models to medical imaging tasks without task-specific fine-tuning. The method emphasizes efficiency by combining test-time augmentations, automated hyperparameter selection via Bayesian optimization, and prompt-refinement strategies to steer predictions. The contribution targets two core challenges: (1) refining the outputs of a pre-trained model and (2) adapting that model to the target domain through an automated, data-driven procedure. The authors report competitive—and in several cases state-of-the-art—results, supported by ablations and validation strategies integrated into the pipeline.

**Strengths:**

- Clear motivation and solid empirical support: The manuscript is generally well written and easy to follow. The identified problem is timely, and the proposed approach achieves competitive (often SOTA) performance according to the reported experiments.
- Ingenious combination of known components: Test-time augmentation, AutoML/Bayesian optimization, and prompt refinement are combined to address complementary facets of the problem in a thoughtful way.
- Automation and practicality: The pipeline is fully automated, including a validation mechanism and domain adaptation procedure, which enhances usability in real-world settings where fine-tuning may be impractical.

**Weaknesses:**

- Limited exposition of module interactions: While each component is described, the paper would benefit from a clearer explanation—ideally a schematic—of how the modules interact end to end (prompting - refinement - TTA - Bayesian optimization). Empirical analysis of interdependencies is also missing. In particular, please assess robustness when upstream (pre-trained) elements underperform: How sensitive is the pipeline to weaker grounding models or to degradation/failures in the refinement block?
- Variance and robustness reporting: Medical imaging results can be highly seed-dependent. Reporting mean ± standard deviation over multiple runs (e.g., 3–5 seeds) for key tables would help assess robustness. If feasible, include variance under common sources of distribution shift (e.g., scanner/site differences).
- The efficiency claim would be stronger with clearer accounting of computational cost (e.g., TTA overhead, Bayesian optimization budget, wall-clock time and GPU hours) relative to fine-tuning baselines at matched accuracy.
- The main contribution lies in the orchestration of established components (prompting, TTA, Bayesian optimization), while the paper’s genuinely new additions appear limited; please better isolate, justify, and quantify the novel parts beyond the integration engineering.

**Questions:**

1. How do performance gains decompose across prompt refinement, TTA, and Bayesian optimization when combined versus in isolation? Any observed synergistic or antagonistic effects?
2. How sensitive are results to prompt wording/length and the number of prompts? Is there an automated prompt search, or are prompts seeded from domain knowledge?
3. How does the approach generalize across different pre-trained backbones or foundation models? Is there a degradation when swapping to weaker models?
4. How does the pipeline handle cross-site or cross-scanner generalization without fine-tuning? Have you tested adaptation using only unlabeled target data from a new site?

---

> ### Author Response · Authors · 2025-11-22
> **Response to Reviewer t9YK (part 1)**
>
> We thank the reviewer for the the positive assessment and constructive feedbacks of our paper. Here we address the raised concerns and questions in detail.
>
> **[W1]** Limited exposition of module interactions.
>
> We have some analyses regarding the sensitivity to module choices (some results are in Appendix due to page limitation). Specifically, we have the following observations which correspond to your specific concerns:
>
> _Sensitivity to weaker grounding model._ Our pipeline is sensitive to the grounding performance since it provides the box prompt to SAM, which is also why the cooperation of LTAs, the Proxy Validator, and BO is essential to our system. Our ablation study in Table 2 shows that the overall performance degrades substantially when ablating the optimal LTAs before CogVLM. Our modular analysis in Table 6 (Appendix H) further shows that replacing CogVLM with lightweight GroundingDINO obviously affects overall performance. On the other hand, using more powerful foundation model components consistently improves accuracy; for example, replacing SAM with MedSAM increases the Dice scores. We report SAM-based results in the main paper for fair comparison with baselines that also use SAM.
>
> _Sensitivity to degradation/failure of the refinement block._
> Compared to the grounding module, our pipeline is less sensitive to the refinement block, since the point prompt is used to help refine the make boundary instead of initial localizaiton. Specifically, we have the following experimental observations to explain the flexibility and contribution of the refinement block:
>
> 1) Our pipeline has a mechanism to actively degrade or even ablate the refinement module if it does not benefit the overall process during BO. This is because the number of boosted points is a hyperparameter optimized by BO. The illustrative results in Figure 7 show that the optimal number of points is learned to be 0 for images where no representative points can be selected to enhance interactive segmentation (e.g., for REFUGE and PROMISE).
>
> 2) By removing the learned optimal refinement block, the accuracy decreases noticeably although not as significantly as degrading the grounding module. For example, the Dice score on BUSI decreases from 61.65 to 59.60 when ablating the refinement step.
>
> ----------------------------------------------------------------------------------------------
>
> **[W2]** Variance and robustness reporting.
>
> Thanks for your suggestion. We currently have the standard deviation numbers (over 3 seeds) for the following datasets due to time limitations, and we will add the remaining standard deviation numbers to the updated paper later. The initial results show that our method is not quite sensitive to random seeds, because we mostly rely on frozen pre-trained models, and randomness is only involves in the BO process which optimizes 20 LTA variables.
>
> |Kvasir|Busi|REFUGE|SkinCancer|
> |--|--|--|--|
> |73.95±1.55|61.68±1.20|79.95±0.52|85.15±1.08|
>
> Regarding the concern about cross-scanner/site generalization, we put more detailed explainations in response to Q4.
>
> ----------------------------------------------------------------------------------------------
>
> **[W3]** Detailed computational cost.
>
> Thank you for your suggestion. The overall computational cost can be discussed from two aspects, which are the _TTA cost_ and _inference cost_. Specifically, the TTA cost scales linearly with two factors: the number of BO iterations and the cost of each BO iteration, which corresponds to the inference time on the whole test set. In contrast, the inference cost is only determined by the sequential foundation models (the LTA cost is trivial) and is no longer affected by the BO process.
>
> Regarding the comparison against a fine-tuning baseline with matched accuracy, we believe that the comparison primarily concerns the TTA cost, since TTA and fine-tuning are different training techniques. Specifically, we compared LoRA (rank = 8) on pre-trained SAM under two settings, which are full training data and 10% training data. We report the Dice scores and training time (hours) below. Our method is slightly heavier than the full-data LoRA in terms of TTA cost. This is reasonable because LoRA is a supervised learning method that achieves optimization using ground-truth labels, whereas zero-shot pipelines require more exploration with proxy validation using only unlabeled test data. The TTA efficiency could be further improved by warm-starting strategies you suggested.
>
> On the other hand, although low-data LoRA is much more efficient due to fewer training examples, it easily reaches a performance upper bound due to data limitations, meaning that further increasing the number of training epochs will not bring continuous accuracy gains.
>
> |Data|AutoMiSeg (zero-shot) |LoRA (full data, 15 epochs)| LoRA (10% data, 30 epochs)|
>  |--|--|--|--|
>  |Kvasir|74.80/4.9hrs|81.38/3.2hrs|66.59/0.6hrs|
>  |Busi|61.65/4.9hrs|72.31/2.1hrs|54.71/0.4hrs|

---

> ### Author Response · Authors · 2025-11-22
> **Response to Reviewer t9YK (part 2)**
>
> **[W4]** Unique contributions of this paper.
>
> Thank you for pointing out this representation issue. We wanted to clarify that despite our pipeline is based on basic model orchestration, the core contribution of our work lies in designing a novel and practical TTA solution, which bridges the gap between the foundation models and specific medical imaging domains. This issue cannot be addressed by simply integrating existing models or applying traditional TTA techniques. Prior studies such as SaLIP and MedCLIP-SAM explored model orchestration and demonstrated promising results on relatively narrow domains, but they did not address the more fundamental challenge of the gap between generic foundation models and diverse medical domains (Table 1). Our empirical ablation studies (Table 2) further quantatitively justify that our novel TTA design is essential for achieving consistently strong performance, compared to the directly system integration.
>
> In summary, the main significances and contributions of our work can be highlighted from the following aspects:
>
> - Our work is among the first to focus on self-adaptating (rather than directly integrating) a sequence of foundation models in medical imaging settings. Similar efforts have recently emerged in general vision domains and have demonstrated promising results [1].
>
> - We introduce the first TTA-based pipeline for interactive medical image segmentation, addressing the domain gap between specialized medical domains and a foundation model pipeline.
>
> - The core idea of our TTA design lies in the effective collaboration between LTAs, Proxy Validator and BO. It is not only novel, but also specifically developed for non-differentiable foundation model pipelines. It fundamentally differs from traditional TTA methods which apply to standard supervised models.
>
>
> ----------------------------------------------------------------------------------------------
>
>
> **[Q1]** Individual and interactive contributions of prompt refinement, TTA and BO.
>
> We first want to clarify that prompt refinement (part of our LTAs), TTA, and BO are not parallel concepts in our framework. Instead, LTAs, Proxy Validator, and BO are three mandatory components that constitute our TTA design:
>
> 1) Learnable parameters / hyperparameters. For our pipeline, these are the so-called _LTAs_, which contain 20 variables controlling image transformations, grounding text prompts, and the number of points for prompt refinement.
>
> 2) Learning objective. Since ground-truth labels are not available in zero-shot settings, we design the _Proxy Validator_ to provide pseudo supervision.
>
> 3) Optimization strategy. _BO_ is used to efficiently optimize the learnable parameters in the search space.
>
> From a mechanical perspective, LTAs (including the prompt refinement module), Proxy Validator, and BO must work synergistically, i.e., none of the three components can be ablated or isolated. The table below helps illustrate our TTA design using standard supervised learning as a reference.
>
> |Framework|Learnable Parameters|Learning Objective|Optimization Strategy|
> |--|--|--|--|
> |Supervised Learning|model weights|supervised loss (e.g., cross-entropy)|SGD (requires end-to-end differentiable)
> |Our TTA (zero-shot)|LTAs with 20 variables |pseudo signal from Proxy Validator| Bayesian Optimization|
>
> We will improve Section 3.7 to provide clearer explanations about the TTA design, especially the roles and relationships of the three components.
>
>
> ----------------------------------------------------------------------------------------------
>
>
> **[Q2]** Sensitivity to text prompts.
>
> The grounding model is sensitive to both the length and wording of text prompts. Since current grounding models handle short prompts better, we constrain CogVLM prompts to within 20 words. We also observe that wording substantially affects segmentation. As shown below, replacing the optimal prompt with a random one may noticeably change performance on REFUGE:
>
> |Optimal|Random1|Random2|Random3|Random4|
> |--|--|--|--|--|
> |79.78|69.16|67.05|78.10|72.68|
>
>
> To address this, our pipeline does perform automated prompt search rather than relying on manually designed prompts. An LLM generates candidate prompts using only the imaging modality (e.g., X-ray) and target (e.g., lungs), and BO selects the best prompt (the prompt index is an optimized variable). The LLM templates (Appendix D.1) are general and require no external medical knowledge.
>
> The number of text prompts is a hyperparameter. We fix it to 10. Using REFUGE, we find that too few prompts can be suboptimal, while too many may harm BO efficiency:
>
> |n=5|n=10|n=20|n=30|
> |--|--|--|--|
> |76.88|79.78|80.67|79.21|
>
> We believe this is because more prompts enlarge the search space and may distract BO, while too few reduce the chance of selecting a good prompt.

---

> ### Author Response · Authors · 2025-11-22
> **Response to Reviewer t9YK (part 3)**
>
> **[Q3]** Performance with different backbones or foundation models.
>
> Our modular design ensures flexibility with respect to different backbones and foundation models. For example, our modular analysis in Appendix H shows the pipeline performance when replacing the default module choices.
>
> We can expect certain performance degradation when using weaker models, and this is also reflected in the results (Table 6). Intuitively, since our main focus is adapting (rather than fine-tuning) the pre-trained models, the maximum capacity of the pre-trained models naturally serves as a bottleneck, since we cannot extract knowledge or capability that does not already exist in them. On the other hand, the modular design of our pipeline also provides opportunities for future integration of more powerful foundation models.
>
> ----------------------------------------------------------------------------------------------
>
>
> **[Q4]** Cross-site or cross-scanner/site generalization.
>
> The issue of cross-scanner/site differences is usually a concern in traditional supervised learning, where examples from a specific source (a particular scanner/site) are collected, labeled, and then used to train a model. This cross-scanner/site generalization capacity is important for those supervised models, since it is unrealistic to annotate additional examples and re-train the model for every new data source. However, in our zero-shot segmentation pipeline, this does not pose a challenge because the pipeline can be adapted to a new scanner/site in a fully interaction-free manner using only unlabeled test data. In other words, we will perform TTA on each individual scanner/site dataset (with no labeled data or fine-tuning needed) to maximize the segmentation accuracy.

---

### Official Review · Reviewer_2vfY · 2025-10-31

**Soundness:** 3
**Presentation:** 2
**Contribution:** 2
**Rating:** 4
**Confidence:** 4

**Summary:**

This paper presents an automatic zero-shot medical image segmentation pipeline named **AutoMiSeg**, which requires neither manual annotations nor human prompts during inference.  The proposed method integrates vision-language models with promptable segmentation models and introduces a novel **Test-Time Adaptation (TTA)** framework.  The core innovation lies in a set of learnable Learnable Test-time Adaptors (LTAs) that align input domains, whose parameters are automatically optimized via Bayesian Optimization, guided by a Proxy Validator that provides feedback for performance improvement.

**Strengths:**

1. This work is likely the first to combine zero-shot segmentation based on foundation models with test-time adaptation, forming a fully automatic and training-free system.

2. The modular architecture is well-structured, achieving a stable compositional zero-shot inference process through a clear task decomposition (Grounding → Prompt Boosting → Segmentation).

3. The experiments cover 7 medical imaging datasets (including fundus, ultrasound, MRI, endoscopy, and dermoscopy), providing comprehensive and comparable results with important ablation studies included.

**Weaknesses:**

1. The core technical novelty is relatively limited, as the main contribution lies in system integration and pipeline optimization, while each module is built upon existing public methods. It is recommended to open-source the full zero-shot segmentation pipeline, which would significantly enhance the paper’s community value.

2. The Bayesian Optimization process involves 100 iterations per dataset, yet the **inference time and computational cost** are not sufficiently reported or discussed.

3. The model’s robustness under **noisy inputs, modality shifts, or grounding failures** has not been evaluated, leaving the discussion on clinical deployment stability incomplete.

4. The related work comparison is insufficient. For instance, recent studies such as **TV-SAM** and other multimodal zero-shot medical segmentation methods share similar ideas and should be explicitly discussed for a fair contextualization.

**Questions:**

1. Could the Bayesian Optimization process be accelerated through **few-shot warm-starting** or **meta-learning** techniques?

2. It is recommended to provide the **correlation statistics between the Proxy Score and the ground-truth Dice** to demonstrate the reliability of the proxy validation mechanism.

3. In a fully automatic segmentation scenario **without human intervention**, how can potential erroneous decisions be prevented from being directly applied in clinical workflows? The authors should further elaborate on **safety strategies** in the Discussion section.

---

> ### Author Response · Authors · 2025-11-22
> **Response to Reviewer 2vfY (part 1)**
>
> We thank the reviewer for the constructive feedbacks of our paper. Here we address the raised concerns and questions in detail.
>
> **[W1]** Technical novelty and significances.
>
> We would like to clarify that the main contribution of our work lies in designing a practical and effective solution to bridge the gap between pre-trained foundation models and specific medical imaging domains. This issue cannot be easily addressed by simply integrating existing models or applying traditional test-time adaptation (TTA) techniques. Prior studies such as SaLIP and MedCLIP-SAM explored this direction and demonstrated promising results on relatively narrow domains, but they did not address the more fundamental challenge of the gap between generic foundation models and diverse medical domains. Our empirical ablation studies (Table 2) further confirm that our novel TTA design is essential for achieving consistently strong performance, compared to the directly system integration.
>
> In summary, the significance of our work can be highlighted from the following aspects:
>
> - Our work is among the first to focus on self-adaptating (rather than directly integrating) a sequence of foundation models in medical imaging settings. Similar efforts have recently emerged in general vision domains and have demonstrated promising results [1].
>
> - We introduce the first TTA-based pipeline for interactive medical image segmentation, addressing the domain gap between specialized medical domains and a foundation model pipeline.
>
> - The core idea of our design lies in the effective collaboration between LTAs, Proxy Validator and BO. It is not only novel, but also specifically developed for non-differentiable foundation model pipelines. It fundamentally differs from traditional TTA methods which apply to standard supervised models.
>
> Our code is provided in the supplementary material, including all the needed configuration files and LLM-generated prompts to ensure reproducibility. The code will be released to github upon acceptance of the paper.
>
> [1] _LLMs Can See and Hear Without Any Training_. ICML 2025.
>
> ----------------------------------------------------------------------------------------------
>
>
> **[W2]** Inference time and computational cost involved by BO.
>
> Bayesian Optimization (BO) is used during the test-time adaptation stage prior to inference, so BO does not affect the inference cost. Similar to other SAM-based segmentation pipelines, the inference time is dominated by computation on the foundation models. We presented the inference time comparison in Appendix G (Table 5). Our inference speed is in the medium range among SAM-based zero-shot pipelines and is only more expensive than MedCLIP-SAM.
>
> The BO process with 100 iterations takes ~5 hours on an NVIDIA A100 GPU, which is comparable to a typical supervised training process (e.g., full fine-tuning or LoRA-based fine-tuning) on SAM. Considering the zero-shot setting (no annotation cost) and our superior segmentation accuracy over the baselines, we believe the computational cost is reasonable.
>
> ----------------------------------------------------------------------------------------------
>
>
> **[W3]** Robustness evaluation on different aspects.
>
> (1) Noisy inputs. We agree that robustness to noisy inputs is important for clinical applicability. Notably, the datasets we evaluate already contain substantial real-world artifacts (e.g., speckle, blur, illumination variations). To further validate robustness under simulated perturbations, we additionally injected Gaussian noise into the test images and re-evaluated AutoMiSeg. As shown below, the performance decreases only moderately under 5% and 20% noise, demonstrating that the pipeline remains stable under considerable noise levels, while the Ultrasound images (Busi) show higher robustness to Gaussian noise than Endoscopy images (Kvasir).
> |Data |clean |5% noise | 20% noise|
> |--|--|--|--|
> |Kvasir|74.80|73.76|63.56|
> |Busi|61.65|61.04|59.45|
>
>
> (2) Modality shifts. Our TTA strategy is explicitly designed to address modality shifts or mismatches. The ablation study confirms that the optimized LTAs are critical for aligning the grounding and segmentation modules to the target domain. Removing these adaptors leads to a dramatic performance drop (Kvasir: 74.80->25.94; BUSI: 61.65->15.72), illustrating the necessity of domain adaptation.
>
> Under TTA frameworks, modality shift is not an issue because adaptation is performed separately for each new modality using unlabeled target images. In other words, we will perform TTA on each individual modality to maximize the segmentation accuracy, rather than applying a pipeline optimized on one modality to another.

---

> ### Author Response · Authors · 2025-11-22
> **Response to Reviewer 2vfY (part 2)**
>
> (3) Grounding failure. We agree that grounding performance is essential to the entire system. We presented empirical results showing that optimal LTAs applied before the grounding model are critical to the pipeline (Section 5.1), and that weaker grounding moddules degrade overall performance (Appendix H). Logically, total grounding failure is indeed fatal to the system, since the SAM segmentation is ultimately generated based on the grounding box. Therefore, the main purpose of the proposed TTA strategy is to prevent such failures by maximizing the grounding capacity through adapted inputs. In addition, replacing the grounding module with more advanced detectors in the future can further reduce grounding errors and enhance overall stability.
>
> ----------------------------------------------------------------------------------------------
>
>
> **[W4]** Insufficient Related work discussion.
>
> We did notice some other studies relevant to our paper. These are not compared either because the code is not published (e.g., TV-SAM) or the task setting is not the same. In terms of technique, TV-SAM relies on a straightforward composition of pre-trained foundation models without addressing the challenge of the domain gap. Thanks to your comments, we will improve the methodological discussions in the Related Work section and update our submission accordingly.
>
> ----------------------------------------------------------------------------------------------
>
>
> **[Q1]** Could the BO process be accelerated?
>
> Thanks for your insightful question. It is indeed feasible to accelerate BO through few-shot warm-starting. Specifically, we use 10% of the test examples for the first 80 iterations as warm-starting, and then use the full test set in the remaining 20 iterations for deeper exploitation. In this way, the BO cost is reduced by more than 70%, while our experiments on REFUGE show that the accuracy remains almost unchanged. We further increased the number of BO iterations using the full test set and observed higher accuracy than the default BO setting, without increasing the overall computation. Therefore, a proper warm-starting (WS) strategy is indeed beneficial for both efficiency and effectiveness.
>
> |BO strategy|Computation Cost|Dice Score|
> |--|--|--|
> |100 iters of full-data (default)| 100%|79.78|
> |80 iters of WS +  20 iters of full-data|28%|79.67|
> |80 iters of WS + 50 iters of full-data|58%|82.50|
>
> ----------------------------------------------------------------------------------------------
>
>
> **[Q2]** Correlation statistics between the proxy score and the groud truth dice.
>
> The correlation experiments are discussed in Section 5.2. We observed clear positive correlations between the normalized proxy score and the ground-truth Dice score (Figure 4, left two plots). We also observed that the proxy score and the ground-truth score exhibit consistent trends during the BO process (Figure 4, right two plots), further confirming their strong positive correlation. We also pointed out that this correlation is only reliable from a statistical perspective, whereas fully relying on each individual validation score as a prediction is risky. For example, Figure 6 shows more than a 15% decrease in true scores when using BO to optimize each individual test example. This also partially explains why the prediction is not reliable if fully relying on image-text alignment for the final decision, as some baseline methods did.
>
> ----------------------------------------------------------------------------------------------
>
>
> **[Q3]** Prevent potential erroneous decisions.
>
> Thanks for the suggestion. We agree that clinical safety is essential. AutoMiSeg is designed as a research prototype rather than a clinical decision-making tool, but we will discuss potential safety concerns and corresponding strategies. Based on our pipeline design, the following strategies can be considered: 1) The validation score can be used as an indicator of prediction uncertainty. 2) We can perform posterior sampling from the BO results to obtain multiple predictions and check their consistency. Since the hyperparameters are mostly related to image transformations, these images can be processed efficiently in parallel. We will update our paper with these discussions accordingly.

---

### Official Review · Reviewer_4xZR · 2025-10-31

**Soundness:** 3
**Presentation:** 3
**Contribution:** 3
**Rating:** 2
**Confidence:** 4

**Summary:**

This paper introduces a fully automatic segmentation pipeline AutoMiSeg for zero-shot medical image segmentation. Specifically, it first uses a large language model (ChatGPT-4o) and a vision-language grounding model (CogVLM) to generate a bounding box from a text description, then refines it with a visual prompt booster (DINOv2), and finally produces a segmentation mask with a promptable model (SAM). It also integrates a test-time adaptation framework that uses Bayesian Optimization to tune a set of Learnable Test-time Adapters (LTAs). Experimental results demonstrate that the proposed method achieves superior segmentation performance.

**Strengths:**

1.	The proposed method is well-motivated in being fully automatic and training-free, thereby enabling zero-shot medical image segmentation.
2.	The authors introduce Learnable Test-time Adaptors (LTAs) and a surrogate validation model to evaluate the segmentation outputs, whose feedback is utilized to optimize the LTAs.

**Weaknesses:**

1.	Limited Novelty: The core technical components of the pipeline are not novel. The paper primarily combines existing techniques: a grounding model (CogVLM/Grounding DINO), a feature-based prompt booster (inspired by CoVP), and a segmenter (SAM) are connected sequentially. The novelty lies in the specific integration and application to this problem, rather than in the invention of new core algorithms.
2.	The evaluation of zero-shot baselines may not be fully comprehensive. Notably, the cited SaLIP model [1] has demonstrated strong performance (e.g., 80%–95% Dice) on modalities like brain MRI and lung CT in its original publication. Its comparatively low performance on the selected datasets in this work (e.g., ~30% Dice on Kvasir and ISIC2016) suggests a potential underutilization of the model or a high sensitivity to specific image domains that warrants further investigation.

[1] Sidra Aleem, Fangyijie Wang, Mayug Maniparambil, Eric Arazo, Julia Dietlmeier, Kathleen Curran, Noel EO’ Connor, and Suzanne Little. Test-time adaptation with salip: A cascade of sam and clip for zero-shot medical image segmentation. In CVPR, 2024.

3.	The supervised models used for comparison in Table 1 (e.g., ResNet-18, ResNet-50) are not representative of state-of-the-art architectures specifically designed for medical image segmentation.
4.	To more robustly validate the advantages of AutoMiSeg, it is recommended to include a wider array of medical imaging modalities (e.g. CT) and anatomical regions. This would provide a more complete and convincing demonstration of the pipeline's generalizability and superior performance across the full spectrum of medical imaging tasks.
5.	Inadequate Justification for the TTA Design: While the proposed Learnable Test-time Adapters (LTAs) are central to the method's performance, the paper provides limited justification for their specific design. There is a lack of comparative experiments with established Test-Time Adaptation (TTA) techniques (e.g., entropy minimization, feature alignment) to demonstrate that the chosen Bayesian Optimization over image transformations is a superior or even competitive strategy. This omission makes it difficult to assess the true contribution and novelty of the LTA module.
6.	Furthermore, the architectural integration of the LTAs within the pipeline remains unclear. For better clarity and reproducibility, it is strongly recommended to include a detailed structural diagram of the LTAs in Figure 2, explicitly illustrating how the transformations are applied and how the hyperparameters are optimized and fed into the different modules.

**Questions:**

please see the weakness

---

> ### Author Response · Authors · 2025-11-22
> **Response to Reviewer 4xZR (part 1)**
>
> We thank the reviewer for the constructive feedbacks of our paper. Here we address the raised concerns and questions in detail.
>
> **[W1]** Limited Novelty.
>
> We would like to clarify that the main contribution of our work lies in designing a practical and effective solution to bridge the gap between pre-trained foundation models and specific medical imaging domains. This issue cannot be easily addressed by simply integrating existing models or applying traditional test-time adaptation (TTA) techniques (more explainations in our response to W5). Prior studies such as SaLIP and MedCLIP-SAM explored this direction and demonstrated promising results on relatively narrow domains, but they did not address the more fundamental challenge of the gap between generic foundation models and diverse medical domains. Our empirical ablation studies (Table 2) confirm that a direct integration is far from satisfactory on various medical tasks, whereas our specific TTA design is essential for ensuring strong zero-shot performance.
>
> In summary, the significance of our work can be highlighted from the following aspects:
>
> - Our work is among the first to focus on self-adaptating (rather than directly integrating) a sequence of foundation models in medical imaging settings. Similar efforts have recently emerged in general vision domains and have demonstrated promising results [1].
>
> - We introduce the first TTA-based pipeline for interactive medical image segmentation, addressing the domain gap between specialized medical domains and a foundation model pipeline.
>
> - The core idea of our design lies in the effective collaboration between LTAs, Proxy Validator and Bayesian Optimization (BO). It is not only novel, but also specifically developed for non-differentiable foundation model pipelines. It fundamentally differs from traditional TTA methods which apply to standard supervised models.
>
> [1] _LLMs Can See and Hear Without Any Training_. ICML 2025.
>
>
> ----------------------------------------------------------------------------------------------
>
>
>
> **[W2]** The evaluation of SaLIP.
>
> We reproduced SaLIP using the released code and achieved strong performance on brain MRI and lung X-ray (not lung CT). The remaining gap between our reproduction and the reported results is likely due to data processing details. For example, we obtained reasonable performance only after converting the float-valued masks to binary labels, which required thresholding. The code also applied rotations, padding, and edge-slice removal. We followed all known steps, but additional preprocessing steps may have been used.
>
> Nevertheless, we evaluated SaLIP and our AutoMiSeg on exactly the same processed brain MRI (GE1.5) and lung X-ray examples. Results in the table below show that SaLIP performs well on these tasks, and our method still demonstrates clear advantages.
>
> |Method | Brain MRI |Lung X-ray|
> |--|--|--|
> |SaLIP|71.37|73.16|
> |AutoMiSeg|78.12|77.92|
>
> From a data perspective, the reported high performance is possibly because the brain/lung segmentation tasks in SaLIP are relatively easy. The segmentation objects in these tasks are generally homogeneous, large-sized, and have clear boundaries (as shown in Figures 3, 4, 7, 8, 9 of the SaLIP paper). In contrast, the datasets we initially used are much more challenging. As illustrated in the examples in Figure 7 of our paper, several datasets (Kvasir, Busi, Promise) contain subtle boundaries.
>
> From a methodological perspective, we believe the following key differences explain why our method is superior to SaLIP across a wider range of medical imaging tasks:
>
> 1) SaLIP directly processes images with SAM and thus suffers from domain gaps between the images and the frozen SAM. When the object boundaries are not naturally clear to SAM, it will fail at the candidate generation stage, leading to total failure in the following segmentation.
>
> 2) Although SaLIP involves CLIP to rank candidate masks according to image-text alignment, it directly relies on CLIP for the final prediction without further validation. This is risky if the two modalities do not align well for an individual example. Our paper illustrated this risk in Figure 6. In contrast, our pipeline employs CLIP as a validator, i.e., the proxy score is more statistical (calculated over a batch of examples) and is used to optimize the LTA variables instead of making decisions.
>
> 3) SaLIP employs a random search on five random images to determine the important hyperparameters for SAM mask generation. This method might be effective for homogeneous targets such as organ recognition. However, for heterogeneous targets such as tumor/lesion recognition (i.e., the breast tumors or skin cancers are with varied sizes and shapes), the few selected training examples may not be representative.

---

> ### Author Response · Authors · 2025-11-22
> **Response to Reviewer 4xZR (part 2)**
>
> **[W3]** Supervised baselines.
>
> The supervised baselines are from the MedSegBench paper (Nature 2024). We are sorry for the potential misleading, but we want to clarify that the ResNet-18/50 actually represent U-Net models with corresponding ResNet backbones, which still serve as very popular and strong baselines for medical image segmentation.
>
> We agree that more powerful architectures such as Swin-UNETR are the state-of-the-art medical models. In this paper, since our main target is to design a zero-shot and training-free pipeline, the performance of supervised models is only used as a reference and as upper bounds.
>
>
> ----------------------------------------------------------------------------------------------
>
>
> **[W4]** Addtional experiments with more image modalities and anatomical regions.
>
> We added new experiments to increase the coverage of imaging modalities and anatomical regions. The newly incorporated datasets include: cc359-GE1.5 (MRI, _brain_), Covid19 (_X-ray_, _lungs_), CHAOS (_CT_, _liver_), wbc (_Microscopy_, _nucleus_), yeaz (_Microscopy_, _cells_) and Robotool (Endoscopy, _surgical tools_). The imaging modalities and anatomical regions noted as _italics_ indicate new coverage. Results shown in the following table confirm that our pipeline consistently works well across a wider spectrum of medical imaging tasks, i.e., significantly outperforming the competitive baseline MedCLIP-SAM and comparable to weak-prompted interactive SAM (WP-SAM).
>
> | Method | GE1.5 | Covid19|CHAOS | wbc |yeaz | Robotool|Average|
> |--|--|--|--|--|--|--|--|
> |SP-SAM (upper bound)|94.42|88.31|91.20|89.52|92.56|86.00|90.34|
> | WP-SAM |87.92 |73.94 |71.70 |81.09|89.25| 76.03|79.99|
> | MedCLIP-SAM|70.02|43.61|62.96|67.24|59.76|50.37|59.33|
> |AutoMiSeg (ours)|78.12|77.92|72.14|76.23|88.01|76.90|78.89|
>
> Due to time and resource constraints, more detailed dataset descriptions and full baseline comparisons will be included in the updated version of the paper later.
>
>
> ----------------------------------------------------------------------------------------------
>
>
> **[W5]** Justification for the TTA design.
>
> We would like to clarify that the traditional TTA techniques do not directly apply to our pipeline because they require parameter updates via gradient backpropagation, e.g., updating BatchNorm/LayerNorm layers or other lightweight adaptors/modules. However, our work aims to automate the SAM-based interactive segmentation framework, which is not end-to-end differentiable (e.g., SAM output is not differentiable w.r.t. the box prompt). Beyond this necessary feasibility constraint, our TTA design has the following additional rationale and advantages:
>
> 1) Our pipeline does not assume updates or even access to the foundation models. In other words, the foundation models can be used through APIs (which is common nowadays) instead of local deployment. Therefore, instead of updating the models, we choose to augment the images to adapt the frozen pre-trained foundation models. This design also helps reduce the risk of over-fitting.
>
> 2) While traditional TTA methods typically adopt entropy minimization or feature alignment as the optimization objective, we use a proxy validator to provide pseudo supervision, which is more informative and aligned with the semantic target.
>
> 3) Traditional TTA methods involve far more learnable parameters than our method. For example, updating only the LayerNorm layers in a ViT-based model introduces over 30k learnable parameters (24 layers × 768 dimensions × 2 parameters per dimension). This requires sufficient test data to produce reliable adaptation statistics. In contrast, our LTAs involve only 20 learnable parameters, which makes our approach more robust when the test set is small.
>
> ----------------------------------------------------------------------------------------------
>
>
> **[W6]** Illustrate the architectural integration of the LTAs.
>
> Thank you for the suggestion. We will add this improvement in our updated paper later.

---

### Official Review · Reviewer_kPfd · 2025-10-31

**Soundness:** 4
**Presentation:** 4
**Contribution:** 4
**Rating:** 6
**Confidence:** 5

**Summary:**

This paper introduces AutoMiSeg, a novel pipeline for fully automatic, zero-shot medical image segmentation. The pipeline compositionally leverages existing foundation models: a vision-language model grounds the target from a text description, a prompt booster refines the input, and a segment-anything model generates the mask. A key contribution is the introduction of a test-time adaptation framework to address the domain gap while applying those general models on medical images, which uses Learnable Test-time Adaptors (LTAs). Their hyperparameters are optimized via Bayesian Optimization, guided by a cleverly designed proxy validator that assesses segmentation quality without ground truth.

**Strengths:**

1.	The method is extensively evaluated on seven diverse medical imaging datasets, where it substantially outperforms all existing zero-shot baselines and achieves competitive performance with weakly-supervised interactive models.
2.	The Bayesian Optimization (BO) framework for Test-Time Adaptation is practical and efficient. It is guided by the custom proxy validator, whose score is demonstrated to have a strong correlation with the true segmentation performance (as measured by the Dice score) making the adaptation both target-aware and effective.
3. The BO strategy is data-efficient. It requires only a minimal number of objective evaluations (on a small, unlabeled subset of the target data) to converge to a robust set of hyperparameters for the entire dataset. This low sample complexity is critical for adapting to a new imaging modality or clinical site without the need for extensive data collection or costly retraining.
4. Instead of the conventional approach of fine-tuning the massive foundation models themselves—a process that demands substantial labeled data and immense computational resources—this work shifts the focus to bridging the domain gap at the input level. This design ensures future-proofing and seamless compatibility with model evolution. The pipeline can directly and immediately benefit from any future improvements in these general-purpose models (e.g., a more powerful SAM 3.0 or a more grounded VLM). Simply plugging in a better base model would likely boost AutoMiSeg's performance without any change to its core adaptation mechanism.

**Weaknesses:**

1. A notable methodological limitation lies in the design of the Proxy Validator. The validation process relies on evaluating the segmented region in isolation by masking out the background. While this is an effective proxy in many scenarios, it overlooks a fundamental aspect of medical image interpretation: context can be critical for accurate identification. By isolating the candidate region, the validator discards this contextual information. Although the results demonstrate a strong correlation between the validator's score and the Dice coefficient, this weakness may limit the framework's robustness and accuracy for certain tasks where contextual reasoning is paramount, potentially causing optimization failures in these edge cases.

**Questions:**

1. Do you have any experiment results for comparison with LoRa or other peft models?

---

> ### Author Response · Authors · 2025-11-22
> **Response to Reviewer kPfd**
>
> We thank the reviewer for the careful reading and the positive assessment of our paper. Here we address the raised concerns (weaknesses) and questions in detail.
>
> **[W1]** Methodological limitation of the Proxy Validator.
>
> Thank you for your insightful observation regarding the methodological limitation of the Proxy Validator. We agree that this limitation is rooted in the fundamental task property of segmentation: because segmentation is a pixel-level prediction task, the validator needs to assess the semantics of a local region (the candidate) in contrast to the remaining pixels. This naturally leads us to validate the segmented region in isolation.
>
> To address part of this risk in our current implementation, we already incorporate a simple but practical strategy. Specifically, for candidate regions whose area is relatively small (less than 50% of the image area), we generate a rectangle that fully covers the candidate mask, and this rectangle is used as the Validator’s input. This design is intended to avoid cases where a candidate region is too small for the Validator to recognize when presented in isolation. Through this rectangle, partial contextual information is included to help the Validator better understand the candidate region. Of course, we admit that this strategy only considers local context and does not fully leverage global contextual information.
>
> Motivated by the reviewer’s valuable suggestion, we plan to further improve the validation module by more systematically incorporating contextual information. One possible direction is to apply explainable AI techniques on CLIP-based models. We noticed relevant ideas applied to weakly-supervised segmentation in general computer vision [1]. We also believe that region-aware CLIP models could further improve the alignment between text features and patch-level image features [2]. However, this goal is still challenging for medical domains due to the lack of high-quality region-aware CLIP models, and we hope to explore this direction in future work.
>
> We will add these discussion into our updated paper later.
>
> [1] Exploring CLIP’s Dense Knowledge for Weakly Supervised Semantic Segmentation. (CVPR'25)
>
> [2] Alpha-CLIP: A CLIP Model Focusing on Wherever You Want. (CVPR'24)
>
> ----------------------------------------------------------------------------------------------
>
> **[Q1]** Comparison with LoRA or other PEFT models.
>
> PEFT methods such as LoRA (in the context of fine-tuning) essentially belong to the supervised learning category, as ground-truth labels are required for model training. Nevertheless, we evaluated LoRA on two representative datasets Kvasir and BUSI. Specifically, we still use the pre-trained SAM as the segmentation model and fine-tune SAM with LoRA (rank = 8), resulting in 0.53% of parameters being trainable. For LoRA, we consider two settings, where we use the full training set and 10% of the training examples, respectively.
>
> The results are shown as follows. As a supervised learning method, the performance of LoRA is largely affected by the number of training examples. When using the full training data (700 examples for Kvasir and 452 examples for Busi), LoRA outperforms our zero-shot AutoMiSeg. However, in low-data scenarios, e.g., using only 10% training data (70 examples for Kvasir and 45 examples for Busi), LoRA does not show superiority over our zero-shot pipeline. This indicates the limitation of supervised learning (including PEFT methods like LoRA) which is highly data-hungry, whereas our zero-shot pipeline does not require any training data.
>
> |Data|AutoMiSeg (zero-shot) |LoRA (full data, 15 epochs)| LoRA (10% data, 30 epochs)|
>  |--|--|--|--|
>  |Kvasir|74.80|81.38|66.59|
>  |Busi|61.65|72.31|54.71|

---

> > ### Comment · Reviewer_kPfd · 2025-11-28
> >
> > Thank you for the response, I have no further question and will maintain my positive rating for now.

---

### Comment · Area_Chair_yEQW · 2025-11-25
**Discussions started**

Dear reviewers,

The authors have provided responses to the reviews. Please read these responses and discuss with authors if you have any further comments/questions.

Best,

AC

---

### Author Response · Authors · 2025-12-03
**Paper updated**

We sincerely thank all reviewers for the thorough evaluation of our submission and for the constructive comments to help improve the clarity and completeness of this work. We carefully addressed all concerns by adding new experiments that broaden the coverage of imaging modalities and anatomical regions, refining our methodological analysis, and clarifying the technical contributions of our work. We believe these revisions strengthen the paper and more clearly demonstrate the significance of our TTA-based zero-shot interactive segmentation pipeline.

We are grateful for the valuable insights provided and hope that the updated version meets the expectations of the reviewers and the AC.

---

### Note · Program_Chairs · 2026-01-17
**Submission Desk Rejected by Program Chairs**

The following references in this submission do not refer to real documents and/or have major errors in bibliographic information:

 Panyu Deng, Honghui Li, Jili He, et al., “Segment Anything Model (SAM) for Medical Image Segmentation: A Survey,” arXiv:2310.10617